

# Prefrontal cortical activation measured by fNIRS during walking: effects of age, disease and secondary task

Paulo H.S. Pelicioni[1,2], Mylou Tijsma[3], Stephen R. Lord[1,2] and Jasmine Menant[1,2]

[1] Falls, Balance and Injury Research Centre, Neuroscience Research Australia, Sydney, NSW, Australia
[2] School of Public Health and Community Medicine, University of New South Wales, Sydney, NSW, Australia
[3] Catharina Hospital, Eindhoven, Netherlands

## ABSTRACT

**Background**. Cognitive processes are required during walking to appropriately respond to environmental and task demands. There are now many studies that have used functional Near-Infrared Spectroscopy (fNIRS) to record brain activation to investigate neural bases of cognitive contributions in gait. The aim of this systematic review was to summarize the published research regarding Prefrontal cortical (PFC) activation patterns during simple and complex walking tasks in young adults, older adults and clinical groups with balance disorders using fNIRS. Our secondary aim was to evaluate each included study based on methodological reporting criteria important for good data quality.

**Methods**. We conducted searches in June 2018 using four databases: Embase, PubMed, Scopus and PsycINFO. The strategy search used was: (((((near infrared spectroscopy) OR functional near infrared spectroscopy) OR nirs) OR fnirs) AND (((gait) OR walking) OR locomotion) AND (((((young) OR adult) OR older) OR elderly) NOT children)) AND (((Brain) OR cortex) OR cortical) for our search. The papers included met the specific review criteria: (i) used fNIRS to measure PFC activation patterns; (ii) included walking tasks (simple and complex) and; (iii) assessed young people, older people and/or clinical groups with balance disorders.

**Results**. Thirty five (describing 75 brain activation comparisons) of the 308 studies retrieved through our search met the inclusion criteria. Based on 6 methodological reporting considerations, 20 were of high quality, 10 were of medium quality and 5 were of low quality. Eleven/20 comparisons in young people, 23/37 comparisons in older people and 15/18 comparisons in clinical groups reported increased PFC activation with increased walking task complexity. The majority of comparisons that used verbal fluency, counting backwards or secondary motor tasks reported increases in PFC activation (83%, 64% and 58% of these studies, respectively). In contrast, no studies found secondary visual tasks increased PFC activation.

**Conclusion**. Increased PFC activation was most common in studies that involved walks comprising secondary verbal fluency and arithmetic tasks. Clinical groups generally showed increased PFC activation irrespective of type of secondary task performed during walking which suggests these groups require more attentional resources for safe walking. Systematic review registration number: PROSPERO 2017 - CRD42017059501.

Corresponding author
Jasmine Menant,
j.menant@neura.edu.au

## INTRODUCTION

Walking relies heavily upon coordinated movement controlled by subcortical structures such as the basal ganglia (*Takakusaki, Tomita & Yano, 2008*). However, cognition is also important for locomotor tasks, particularly tasks that require attention and processing speed, such as multi-tasking and gait adaptability (*Montero-Odasso et al., 2012*; *Caetano et al., 2017*). Traditionally, the role of cognition has been assessed using dual-task paradigms (walking while performing a secondary cognitive task) which provide indications of the role of attention and executive function in the regulation of gait control (*Montero-Odasso et al., 2012*) and the negotiation of obstacles (*Caetano et al., 2017*; *Caetano et al., 2018*). Impaired cognitive processing has been associated with reduced gait speed and increased gait variability during complex gait (*Killane et al., 2014*; *Hausdorff, 2005*), however how higher level brain areas are activated during complex walking tasks is still unclear.

Functional near-infrared spectroscopy (fNIRS) is an optical neuroimaging technique for investigating cortical brain area activation while participants move freely. This technique is particularly useful for monitoring hemodynamic responses to brain activation (i.e., changes in oxygenated (oxyHB) and deoxygenated hemoglobin (deoxyHB)) in cortical regions before and after stimulation (i.e., resting followed by simple walking or simple walking followed by dual-task walking) (*Leff et al., 2011*).

Two overlapping theories have been posited for relative changes in cortical activity as measured with fNIRS. The first suggests reduced activity represents decreased use of a brain region and therefore increased efficiency (*Lustig et al., 2009*; *Grady, 2012*). The second suggests increased cortical activity is a compensatory mechanism and reflects over-recruitment and reduced efficiency (*Cabeza et al., 2002*; *Reuter-Lorenz & Cappell, 2008*; *Grady, 2012*).

Several reports of brain activation during walking using fNIRS have been published in the last decade. Activation of the Prefrontal Cortex (PFC) (easily accessible using fNIRS) has often been investigated during walking tasks (*Leff et al., 2011*). Brain motor areas investigated also include the Pre Motor Cortex (PMC), the Pre Supplementary Motor Area (preSMA), the Supplementary Motor Area (SMA) and the Sensory Motor Cortex (SMC) (*Harada et al., 2009*; *Koenraadt et al., 2014*; *Lu et al., 2015*; *Suzuki et al., 2004*; *Suzuki et al., 2008*). Thus, there is now considerable literature that requires synthesizing and systematic review of the main findings related to the brain activation as assessed by fNIRS during walking tasks.

Some recent reviews have examined fNIRS and gait. These reviews have addressed (i) methodological aspects (*Herold et al., 2017*; *Vitorio et al., 2017*); (ii) data processing techniques (*Vitorio et al., 2017*); (iii) or restricted their focus to ageing (*Vitorio et al., 2017*; *Stuart et al., 2018*), Parkinson's disease (PD) and Parkinsonism syndromes (*Vitorio et al., 2017*; *Gramigna et al., 2017*; *Stuart et al., 2018*) or Stroke (*Gramigna et al., 2017*).

Further analysis and synthesis of published fNIRS studies are required to gain a better understanding of (i) brain activation changes during complex walking compared to simple walking or standing; (ii) brain activation patterns in healthy young people as this group provides the model of intact cognitive functioning; and (iii) brain functioning in diverse

clinical groups with walking and neurological impairments. A methodological scale is also required to assist in the evaluation of the literature published to date.

Thus, we conducted a systematic review to summarize the published findings regarding brain activation patterns during simple and complex walking tasks in young adults, older adults and clinical groups with balance disorders, to gain an insight into neural processes required for ambulation. Our primary objectives were to determine whether (i) PFC activation patterns change when people perform gait tasks of increasing complexity requiring concomitant somatosensory, motor or cognitive tasks; (ii) PFC activation patterns during gait differ between young and older people and between patient groups and healthy controls. Our secondary aim was to evaluate each included study based on six methodological reporting criteria important for good data quality.

## METHODOLOGY

### Search strategy

We followed the Preferred Reporting Items for Systematic Reviews and Meta-Analyses (PRISMA) statements defined by *Moher et al. (2009)* to identify and screen the articles included in this systematic review. We conducted searches in June 2018 using four databases: Embase, PubMed, Scopus and PsycINFO. A protocol was prospectively registered with the International Prospective Register of Systematic Reviews (PROSPERO) (registration number: PROSPERO 2017: CRD42017059501). We used the following Booleans terms: (((((near infrared spectroscopy) OR functional near infrared spectroscopy) OR nirs) OR fnirs) AND (((gait) OR walking) OR locomotion) AND (((((young) OR adult) OR older) OR elderly) NOT children)) AND (((Brain) OR cortex) OR cortical) for our search. We considered papers in English, Portuguese, Dutch and French.

### Selection criteria

Study identification and screening were conducted independently by PP and MT or PP and JM with disagreements resolved by consultation and input from a third researcher (JM or SL). At stage 1 (identification), the researchers screened the manuscript titles and selected those that were consistent with the broad inclusion criteria. Studies were excluded if: (i) they were not in line with the review objectives; (ii) were conference abstracts with insufficient information for data extraction; (iii) were conducted in animals; (iv) were conducted in children/infants; (v) used fNIRS for other purposes (e.g., muscle studies); (vi) used fNIRS for standalone purposes (i.e., no walking assessment); (vii) used a device other than fNIRS (e.g., electroencephalogram) and/or; (viii) were published in a language other than English, Portuguese, Dutch or French.

At stage 2 (screening), the researchers screened the abstracts to identify papers that met the other specific review criteria: (i) used fNIRS to measure PFC cortical activation patterns; (ii) included walking tasks (simple and complex) other than stepping and; (iii) assessed young people, older people and/or clinical groups with balance disorders (defined as any peripheral or neurological condition that affects balance control). At stage 3 (eligibility), the full-text articles were assessed for eligibility. A manual search for additional relevant references from published reviews and articles was also conducted at this point. Articles

were further excluded if: (i) they did not include a walking analysis; (ii) gait tasks involved walking speeds slower than 3 km/h; (iii) the participants performed tasks other than walking (i.e., stepping tasks); (iv) there was no baseline data comparison. Papers meeting all selection criteria were included at stage 4 (included papers) and relevant information was extracted from the papers by three authors (PP, MT and JM).

The primary outcome of the review was PFC activity change post stimulation. This was operationalized by changes in oxyHb, deoxyHb (gold standard measurement in brain magnetic resonance imaging (*Obrig & Villringer, 2003*; *Lindauer et al., 2010*), tissue oxygenation index (ratio of oxygenated to total tissue hemoglobin) and total hemoglobin level (sum of both oxyHb and deoxyHb). All hemodynamic changes reported in this review reflect statistically significant results reported by the authors of each study, i.e., $p$ values <0.5.

## Data extraction

From the included studies, relevant data were extracted and summarized for further analysis (Table 1). These included: (i) author and year; (ii) sample characteristics; (iii) study aims; (iv) gait assessment; (v) secondary task types; (vi) equipment details; (vii) fNIRS parameters used to describe the brain activation; (viii) control of motion artefacts and filtering; (ix) main findings; (x) study limitations; and (xi) conclusions.

A methodological reporting scale based on availability of information provided in the papers was devised. It comprised one point for the following 6 items: (i) equipment details described adequately, i.e., number of channels for the fNIRS, optode distances; (ii) movement artefacts/high frequency noise controlled for; (iii) use of either the 10–5, 10–10 or 10–20 electroencephalography electrode system to guide the optode placement; (iv) interference with external light controlled for; (v) heart rate changes and physiological noises controlled for; (vi) sample size in each group >10. Papers were classified as follows: low methodological reporting quality: total score $\leq$ 2 points; medium methodological reporting quality: total score 3 and 4 points; high methodological reporting quality: total score $\geq$ 5 points.

## RESULTS

A total of 308 study records were identified from the four databases; 103 unique studies with the removal of duplicates. The manual search of the references of these studies identified 12 further relevant studies. Of these 115 studies, 58 were deemed eligible for full-text assessment based on abstract review, with 35 meeting our final inclusion criteria (Fig. 1). The data extracted from these studies are summarized in Table 1. Nine studies involved young adults only (simple walking (1), fast walking (1), motor task (2), motor and cognitive tasks (2) and only cognitive tasks (3)); 4 studies involved young and older people (motor and cognitive tasks (1), only cognitive tasks (3)); 1 study involved young and older people and a clinical group with balance disorders (stroke survivors) performing simple walking, motor and cognitive tasks; 10 studies involved older people only (fast walking (2), motor and cognitive tasks (1), somatosensory and cognitive tasks (1), somatosensory and motor and cognitive tasks (1), only cognitive tasks (5)); 8 studies involved older people

 

**Table 1  Summary of fNIRS studies.**

| Author, year | Sample | Aims | Gait assessment | Secondary task | Equipment details | Measurable parameters | Brain areas | Controlling, artefacts and filtering | Results | Limitations | Conclusion |
|---|---|---|---|---|---|---|---|---|---|---|---|
| *Al-Yahya et al. (2016)* | 19 individuals with chronic stroke, 60 ± 15 years old, 2 women; 20 healthy controls, 54 ± 9 years old, 8 women. | To investigate PFC activation and relationships between PFC activation and gait measures while walking under single-task and DT condition in individuals with stroke and healthy controls. | Walking on a treadmill at self-selected speed (5 30s trials). Gait spatiotemporal parameters estimated with inverted pendulum model using kinematic data from inertial sensor attached at the level of the fourth lumbar vertebra (close to the centre of mass). Additional single task: counting while standing Baseline: unclear. | Counting backward in 7 from a random number between 291 and 299 while walking (5 trials) for 30s. The outcomes were: rate and accuracy of correct answers. No advice given as to which task to prioritize during DT-walking. | Oxymon Mk III system, (Artinis Medical Systems). Wavelengths: 782 and 859 nm. 8 channels. Interoptode position 3.0 cm. Sampling rate: 10 Hz. | OxyHb and deoxyHb analysed during 10s window between 6 and 16s post stimulus onset. | PFC. | Blood pressure was measured at the beginning and after the end of each trial. To remove high-frequency noise (cardiac pulsation) fNIRS signals were then low-pass filtered at 0.67 Hz cut-off frequency. | Increased oxyHb concentration and decreased deoxyHb in DT walking compared with simple-task walking and with standing while counting for both groups and hemispheres. | The authors pointed the sample size as a limitation. | Increased PFC activation in DT walking versus single task, among stroke patients. No between-group differences in PFC activation during walking. |
| *Beurskens et al. (2014)* | 15 young adults, 25 ± 3 years; 10 HOA, 71 ± 4 years old. | To compare the effects of completing a secondary visual checking task versus a verbal memory task during walking on PFC activation in young and HOA. | Walking on a treadmill at self-selected speed (2 30s trials per condition). Baseline condition: rest period seating on a chair. Gait outcomes: step duration, step length and number of steps. | (i) visual checking (seated); (ii) alphabet recall (seated); (iii) walking and visual checking; (iv) walking and alphabet recall. All tasks duration: 30s. Secondary task outcomes: number of checked boxes per second (visual checking) and number of correctly recited letters per second (alphabet recall). | DYNOT Imaging System, NIRx Medical Technologies, LLC. Wavelengths: 760 nm and 830 nm. 14 channels. Interoptode position: 2.2 and 2.5 cm. Sampling rate: 1.81 Hz. | OxyHb and deoxyHb. | PFC. | Tasks were conducted in a dimly illuminated room. Each channel was visually inspected and movement artefacts were corrected and data were reconstructed. Hemodynamic response function low pass filter and wavelet-minimum description length de-trending algorithm to remove possible global trends due to breathing, heartbeat, vasoconstriction or experimental influences. | No between-group difference in brain activation across all channels. No significant effect of verbal memory secondary task (vs. simple walking task) on brain activation in young and HOA. Significant age × condition for visual checking DT: no significant effect of visual checking on hemodynamic response in young whereas significant decrease in PFC activation during DT walking with a complex visual task in HOA. Only one significant correlation between reduced cortical activity in BA 10 and increased DT costs (number of steps) during the DT involving visual checking. | | Likely shift of processing resources from the PFC to other brain regions (not analysed in this study) when HOA faced the challenge of walking and concurrently executing a visually demanding task. |
| *Caliandro et al. (2015)* | 19 individuals with chronic gait ataxia, 31-70 years old, 10 women; 15 healthy controls, 36–73 years old, 8 women. | To investigate whether PFC (BA 10) functioning during ataxic gait is linked to compensatory mechanisms or to the typical intra-subject variability of the ataxic gait. | Overground walking for 10 m at a self-selected speed (10 trials). Baseline: last 10s of upright standing period between each trial once stable fNIRS signal. Kinematic and spatiotemporal gait variables recorded with a motion capture system: stance, swing and double support duration; gait speed; step length and width; lower limb kinematics including temporal intra-subject variability of hip, knee and ankle joints. | None. | NIRO-200, Hamatsu Photonics KK. Wavelengths: 775 nm, 810 nm and 850 nm. 2 channels. Interoptode position: 4 cm. Sampling rate: 2 Hz. | OxyHb, deoxyHb and TOI. | PFC. | To reduce movement artefacts the position of optodes were stabilized fixing them to the head by a double-sided adhesive tape. Velcro was used to reduce the influence of skin blood flow on fNIRS signal. Probe holders covered by a black cloth to avoid infrared light interferences. Two recordings 30min apart to verify influence of infrared system on and off on fNIRS signal. A 0.1 Hz low-pass filter applied to reduce cardiac, breathing signals and low frequency oscillations due to blood pressure. Blood pressure and heart rates recordings before and after motor task. | Significant oxyHb concentration in both channels when walking compared to baseline in patients with ataxia whiles no difference in the controls. No difference in deoxyHb in either group. Positive correlation between increased oxyHB activity of the PFC bilaterally and wider step. No significant correlations between bilateral PFC activation and variability of the joint kinematic parameters. | Small sample size which is acknowledged by the author. Influence of skin blood flow on fNIRS signals especially as interoptode distance is high (4 cm). | Increased PFC activity during walking (versus standing) is associated with increased step width and therefore it might be involved in maintaining compensatory mechanisms rather than be correlated with primary defective cerebellar control. |

**Table 1** (*continued*)

| Author, year | Sample | Aims | Gait assessment | Secondary task | Equipment details | Measurable parameters | Brain areas | Controlling, artefacts and filtering | Results | Limitations | Conclusion |
|---|---|---|---|---|---|---|---|---|---|---|---|
| *Chaparro et al. (2017)* | 10 individuals with MS (mean 56 ± 5 years old, 8 women) and 12 HOA (mean 63 ± 4years, 9 women). | To examine the effects of partial body-weight support—30% body weight support on PFC activation while DT in HOA with and without MS. | One trial per condition (simple and complex walk in non-weight-bearing and partial body-weight support conditions). Self-paced treadmill walking trial consisted of a 30 s warm-up, 30 s period at comfortable self-paced speed and a 15 s period to decelerate. Baseline: Standing still and counting silently in their head, 10 s before each condition. Gait outcomes: gait speed, stride length, stride time and stride frequency. | (i) Walking and talking: recite alternating letters of the alphabet. Outcome: utterance rate (number of correct utterances divided by trial time). | fNIR Imager 1000, (fNIR Devices LLC, Potomac, MD). 16 channels. Interoptode position: 2.5 cm Sampling rate: 2 Hz. | OxyHb. | PFC. | Raw data visually inspected for excessive noise, saturation or dark current conditions. Low-pass filter with a cut-off frequency at 0.14 Hz to remove the physiological effects and any additional noise | MS participants exhibited higher activation patterns in all conditions (i.e., task and body weight support) when compared to HOA. Significantly greater PFC activation in DT compared with single walking task. Task × cohort interaction whereby greater PFC activation level in MS in DT compared to controls. Cohort × support condition interaction whereby controls showed greater activation in partial body-weight support compared with MS. Cohort × task × support condition whereby MS patients showed higher PFC activation in non- body-weight support condition in DT compared with the controls. No significant correlation between gait parameters and oxyHb levels. Similar levels of activation during the last 10s when compared to the first 10s of trials in controls during simple walking in partial body-weight support and in MS during the DT partial body-weight support trials; this indicates maintenance of PFC activation levels across the trial. Significant time × task × support condition × cohort × time interaction suggesting as difficulty increased and partial body-weight support was provided, HOA increased PFC activation across time while MS maintained PFC activation patterns. | No reporting on control of external light, motion artefacts of optode placements. Authors acknowledge small sample sizes and numerous comparisons. | MS patients unable to maintain their PFC activation levels across DT walking condition, unless provided with partial body-weight support. Findings may suggest that the use of partial body-weight support may cause a therapeutic effect, which allows individuals with limitations in physical function to maintain their PFC activation levels. |
| *Chen et al. (2017)* | 90 HOA, 78 ± 5 years old, 46 women. | To examine the changes in PFC oxyHb levels during obstacle negotiation under single and DT conditions in HOA. | Walk on an electronic pathway at their self-selected pace for three consecutive loops. Baseline prior to each trial: 10s standing still and counting silently in the head at a rate of one number per second. Gait outcome: stride velocity. | 1 trial in each condition: (i) simple walking; (ii) walking while talk (instruction to pay equally attention to both tasks; recite alternate letters of alphabet); (iii) obstacle negotiation (virtual pot holes of elliptical shape); (iv) obstacle negotiation during walking while talk (same instructions as for walking while talking). | fNIR Imager 1000, (fNIR Devices LLC, Potomac, MD). 16 channels. Interoptode position: 2.5 cm Sampling rate: 2 Hz. | OxyHb. | PFC. | Data were visually inspected and removed in case of saturation or dark noise. Low-pass filter with a cut-off frequency at 0.14 Hz to eliminate possible respiration and heart rate signals and unwanted high frequency noise. The additional frequency noise was identified and removed by an expert data analyst. | PFC oxyHb levels significantly higher in DT condition vs. single task irrespective of obstacle condition. Slow gait moderated the increase in PFC activation in obstacle conditions: compared to participants with normal gait speed, in those with slow gait, PFC activation levels were significantly increased in obstacle condition relative to simple walking. | No reporting of controlling for external light. The authors also reported that the absence of deoxyHb analysis is a limitation, as well as that other brain areas involved with obstacle negotiation were not analysed. | Individuals with mobility limitations (slow gait) utilized more cognitive resources when navigating around obstacles. |

**Table 1** (*continued*)

| Author, year | Sample | Aims | Gait assessment | Secondary task | Equipment details | Measurable parameters | Brain areas | Controlling, artefacts and filtering | Results | Limitations | Conclusion |
|---|---|---|---|---|---|---|---|---|---|---|---|
| *Clark et al. (2014a)* | 14 HOA, 77 ± 6 years old. | To determine if enhancing somatosensory feedback (with a textured insole) can reduce controlled processing during walking, as assessed by PFC activation. | (i) Participants walked for 100 m (5 consecutive laps around a 20 m course) with a 5.2 m instrumented walkway on the pathway for 60–120s; (ii) participants walked on a treadmill at a self-selected speed. The participants performed 1 trial per task. Baseline: normal walking with normal shoes Gait outcomes: walking speed, step length, double support time, variability of step length and of double support time. | 1 walking trial per condition: (i) barefoot; (ii) own shoes (iii) own shoes with a texture insole (to enhance somatosensory feedback) (iv) with normal shoes performing a verbal fluency task. | NIRO-200, Hamatsu Photonics KK. Interoptode position: 3 cm. | OxyHb, deoxyHb and TOI. | PFC. | None. | Increased right PFC activation for treadmill walking versus overground. Relative to baseline, textured insoles yielded a bilateral reduction of PFC activity for treadmill walking and for overground walking. Relative to baseline, barefoot walking yielded lower PFC for treadmill walking, but not overground walking. Increased bilateral PFC activation during DT was observed only in overground walking and not for walking on treadmill. | The authors did not report any controlling for movement artefacts, blood pressure or heart rate changes. Some missing data meaning that each comparison includes data from at least 11 out of 14 participants. | Enhanced somatosensory feedback reduces PFC activity during walking in HOA. This suggests a less intensive utilization of controlled processing during walking. |
| *Clark et al. (2014b)* | 16 older adults with mild mobility deficits (specific criteria), 77.2 ± 5.6 years old, 8 women. | (i) to assess the extent to which PFC activity and skin conductance level were responsive to Central Nervous System demands in preparation and performance phases of complex walking tasks; (ii) to assess the potential link between PFC activation levels and gait quality during performance of the complex walking tasks. | Walking for 90 m (5 consecutive laps around an 18 m course) over a 5.2 m instrumented walkway (1 trial per condition). Baseline: simple walking. Each trial split into 10s-epoch immediately prior to task start (preparation) and full steady-state walking period (performance). Gait outcomes: spatiotemporal parameters. | (i) verbal fluency task; (ii) Participants walked in a dark room; (iii) Participants carried a tray while walking; (iv) Participant stepped over small obstacles along the walking path; (v) Participants wore an adjustable weighted vest with a load equal to 10% of body weight. Tasks and baseline condition had same duration. No instructions regarding task prioritization. | NIRO-200, Hamatsu Photonics KK. Interoptode position: 3 cm. Sampling rate: 2 Hz. | TOI. | PFC: left and right anterior PFC (BA 10). | Probes placed high on the forehead to avoid temporalis muscle and sufficiently lateral to avoid the superior sagittal sinus. | Main effect of task on TOI during preparation phase: TOI significantly increased compared with control in all tasks but verbal. Significant effect of task on TOI during performance phase: TOI significantly increased in verbal, obstacles and vest tasks. No significant increase of TOI between preparation and performance phases within any task (trend for verbal). High response subgroup (increased PFC activation between control task preparation and complex tasks performance) had less gait disturbance for 76% of the variables. | The authors did not report any controlling for movement artefacts, blood pressure or heart rate changes. | Preparation and performance of complex walking tasks in older adults with mild mobility deficits requires heightened utilization of Central Nervous System resources. |
| *Doi et al. (2013)* | 16 older adults with MCI, 75.4 ± 7.2 years old. | To examine PFC activation during DT walking compared with simple walking in older adults with MCI. To determine if there is a relationship between PFC activation during DT walking and executive function (stroop interference). | Walking at a self-selected pace along a 10m corridor. 3 trials per condition. Each trial/block: 10s standing still, 20s walk, 30s standing still. Back and forth for 30s (3 trials). Baseline: first 10-s pre-walking and final 10-s post walking. Gait outcome: gait speed. | Participants performed a verbal fluency task while walking (3 trials). Task was of equal duration to the gait assessment. Gait outcome assessed was equal to the baseline. | OEG-16 system, Spectratech Inc. Wavelengths: 770 and 840 nm. 16 channel and 12 optodes (6 sources and 6 detectors). Interoptode position 3.0 cm. Sampling rate: 1.54 Hz. | OxyHb. | PFC: right inferior frontal gyrus and left inferior frontal gyrus. | Data signals were filtered with a 0.05 Hz low-pass filter to reject artefacts caused by minor movements of the subject. | Significant increase in oxyHb level during DT walking compared with simple walking. Significant correlation between decreased oxyHb and worse executive function performance. Walking speed was slower during DT walking compared with simple walking. | The authors pointed the small sample size and the heterogeneity of individuals as limitation. | First study to show that PFC activation of older adults with MCI is significantly compared with single walking task. Reduced brain activation during DT walking was correlated with worse executive function. |

**Table 1** (*continued*)

| Author, year | Sample | Aims | Gait assessment | Secondary task | Equipment details | Measurable parameters | Brain areas | Controlling, artefacts and filtering | Results | Limitations | Conclusion |
|---|---|---|---|---|---|---|---|---|---|---|---|
| *Eggenberger et al. (2016)* | 33 HOA (from initial sample of 42 partici­pants), 75 ± 7 years old, 64% female. | To investigate if 8 weeks (3 × 30min /week) of exercise train­ing (interac­tive cognitive­motor video game training or balance and stretching train­ing) ex induces changes in PFC activation levels during challeng­ing treadmill walking (and elicits associated changes in cog­nitive executive functions). | The individuals walked on a tread­mill at preferred and fast walking speed (4 trials per condition). Baseline: very slow walking at 0.2 km/h after each trial for 30s. | Fast speed (complex condi­tion): addition of 2 km/h to preferred gait speed. | Oxiplex TS Tis­sue spectrom­eter. Wave­lengths: 690 and 830 nm. Two sensors. Sam­pling rate: 1 Hz. | OxyHb and deoxyHb. | PFC. | Motion artefacts in oxyHb and deoxyHb were ex­cluded based on speci­fied ranges. Procedure to minimise bias from Mayer waves also applied. Use of multi-distance fNIRS instrument to eliminate measurement bias from skin blood flow changes. | No significant difference between left and right PFC activation at baseline. No significant difference in PFC activation at baseline between fast and preferred walking conditions. | No exter­nal light control was reported. The authors pointed that other brain areas asso­ciated with walking should be assessed in future inter­ventions. | Increased gait speed from pre­ferred to fast while walking on a treadmill did not induce increased PFC activation in HOA. |
| *Harada et al. (2009)* | 15 HOA: (i) high gait capac­ity group, 62.0 ± 3.7 years old ($n = 7$); low gait capacity group, 63.0 ± 3.9 years old ($n = 8$). | To evaluate changes in cor­tical activation patterns during walking at low, moderate, and high speeds and to determine whether gait capacity is as­sociated with regional activa­tion patterns in HOA. | 60 s walking tri­als on a treadmill (OxyHb analysed on 20s after reach­ing target speed) Baseline: last 10s of 40s standing period between trials. Gait outcome: cadence. | Walking speed increase in 3 randomly pre­sented condi­tions: 30%, 50% and 70% (high intensity) in­tensity based on heart rate measurements made during an incremental waking test (3 trials at each speed). | OMM-2001, Shimadzu. Wavelengths: 780, 805 and 830 nm. 42 channels and 28 optodes (12 source and 16 detectors). In­teroptode posi­tion 3.0 cm. Sampling rate 5.26 Hz. | OxyHb. | 7 regions of interest: left and right PFC, left and right PMC, pre-SMA, SMA and medial SMC. | Participants' heart rate was measured during walking and their blood pressure was measured immediately after each walking task. | A greater increase in oxyHb in the left PFC and the SMA during walking at 70% intensity than at 50% or 30%. Increased activa­tion in the medial SMC and SMA was correlated with increased gait speed and cadence. At 70% in­tensity, left PFC activation was greater in low gait ca­pacity group than in high gait capacity group (gait speed >6 km/h at 70% in­tensity). | No control for move­ment arte­facts. | Left PFC is in­volved in the control gait speed; involve­ment of the left PFC might de­pend on an age-related decline in gait capacity in HOA. |
| *Hawkins et al. (2018)* | 24 stroke sur­vivors, 58.0 ± 9.3 years old, 8 women; 15 HOA, 77.2 ± 5.6 years old, 8 women; 9 healthy young adults, 22.4 ± 3.2 years old, 5 women. | To investigate between-group differences in executive con­trol of walking. To investigate the extent to which walking-related PFC activity fits exist­ing cognitive frame-works of the PFC over-activation. | Walking at preferred pace on an 18-m oval-shaped course. The healthy groups: 5 laps; stroke sur­vivors: between 2 and 3 laps. 67s worth of data in each trial split into early and late peri­ods (7–37 s, 37–67 s). Gait outcome: walking speed. Base­line: quiet standing at the start of the course (10s) pre-trial. | (i) Simple walk­ing; (ii) walking over obstacles; (iii) walking with a verbal fluency task. | NIRO-200, Hamatsu Ph­tonics KK. Wavelengths: 735 and 810 nm. 2 channels (each set con­tains one source and one detec­tor). Interop­tode position: 3 cm. Sampling rate: 2 Hz. | OxyHb and deoxyHb. | PFC. | All data were inspected for signal artefact: au­tomatically identified as amplitude offset of the oxyHb signal exceeding 1uM within a 2-second period, and/or a 2-s sliding window standard devia­tion of the oxyHb signal that exceeded 3 standard deviations of the original full signal. | Significant effect of group for PFC activity during simple and obstacle walk­ing tasks, with healthy young group exhibit­ing the lowest level of PFC activity, followed by the HOA group and the stroke survivor group. In young adults vs the 2 other groups, significantly greater remaining PFC ac­tivity during simple walk­ing compared with verbal fluency DT. No significant difference between HOA and stroke survivors in re­maining PFC resources in typical and obstacle condi­tions compared with ver­bal fluency task condition. | No de­scription of optode placement, neither for controlling of exter­nal light and heart changes confound­ing. Finally, the sample size of one group was <10 par­ticipants. Groups walking speeds are different. | Young adults have more re­maining PFC resources for attending to complex walk­ing conditions and/or sec­ondary cogni­tive tasks during walking. There is a heightened use of execu­tive control re­sources in HOA and stroke sur­vivors during walking. The level of PFC resource uti­lization, partic­ularly during complex walk­ing tasks may approach the ceiling of avail­able resources for individu­als who have walking impair­ments. |

**Table 1** (*continued*)

| Author, year | Sample | Aims | Gait assessment | Secondary task | Equipment details | Measurable parameters | Brain areas | Controlling, artefacts and filtering | Results | Limitations | Conclusion |
|---|---|---|---|---|---|---|---|---|---|---|---|
| *Hernandez et al. (2016)* | 8 individuals with MS, 57 ± 5 years old, 6 women and 8 healthy controls, 61 ± 4 years old, 6 women. | To investigate the levels of PFC activation during gait under single and DT conditions in community-dwelling individuals with and without MS using fNIRS. | Walking at normal pace along an electronic pathway at self-selected speed for 45 metres in a continuous looping (1 trial). Baseline: 10s standing trial with participants counting silently in their head. Gait outcome: gait speed. | Walking while talking: Reciting alternate letters of the alphabet while walking. Instruction to pay equal attention to walking and talking. Tasks and baseline condition of equal duration to the gait assessment. The outcomes were: correct letter rate and errors per minute. | Imager 1000, fNIRS Devices LLC. Wavelength: 730, 805 and 850 nm. 16 channel and 16 optodes (4 sources and 10 detectors). Interoptode position: 2.5 cm. Sampling rate: 2 Hz. | OxyHb. | PFC. | The lighting in the test room was controlled for: approximately 150lx or 1/3 of typical office lighting. To eliminate possible respiration, heart rate signals, and unwanted high-frequency noise, raw intensity measurements at 730 and 850 nm were low-pass filtered with a finite impulse response filter with a cut-off frequency of 0.14 Hz. Motion artefacts remaining were eliminated by visual inspection of an expert fNIRS data analyst. | Individuals with MS had greater elevations in PFC oxyHb levels in walking while talking task versus simple walking and compared with healthy controls. Larger increases in oxyHb levels in MS with low disability in comparison to high disability scores during WWT, despite later group walking slower yet not significantly. | The authors pointed the small sample size as a limitation. | Increased PFC activation in DT walk versus simple walk, compared to healthy controls Despite similar gait speed and cognitive performance. |
| *Hill et al. (2013)* | 12 healthy young adults aged 18–22 years old. | To determine whether incremental cognitive workload could be detected both behaviourally and via fNIRS. | Walking for 7.6 m (15 trials). Baseline: 10s standing still. Gait outcome: gait speed. | Half of the participants counted backward by 1 (easy) and the other half of participants counted backwards by 7 (difficult) while walking. Tasks and baseline condition of equal duration to the gait assessment. The outcome was: counting performance. | Imager 1000, fNIR Devices LLC. 16 optodes. | OxyHb. | PFC dorsolateral PFC (BA 9 and 46), anterior PFC (BA 10), and part of inferior frontal gyrus (BA 45). | A low-pass filter was applied to eliminate interference from heart rate, respiration, and other unwanted noise signals. Motion artefact rejection routine applied to eliminate uninterpretable data due to excessive head movement. | Significantly greater oxyHb level in PFC in the difficult versus easy DT walking condition when collapsed across optodes locations. Cortical activation levels in the left-hemisphere were significantly higher in the difficult versus easy condition; only trend was observed for right hemisphere. | Movement artefacts were not controlled in this study, only removed by processing data. | Higher level of PFC activity during a high load condition was observed, which was more pronounced in the left hemisphere than in the right hemisphere. |
| *Holtzer et al. (2011)* | 11 HOA aged 69–88 years old, 7 women; 11 healthy young aged 19–29 years old. | To evaluate whether increased PFC activation would be detected during walking while talking as compared with normal walking; and whether the increase in PFC activation during walking would be greater in young compared with HOA. | Walking on an electronic pathway at self-selected speed for 4.5 m (6 trials). Baseline: 5s standing trial pre-walk. Gait outcome: gait speed. | Reciting alternate letters of the alphabet while walking. Instruction to pay equal attention to their walking and talking. Outcomes: letter rate and errors per minute. | Sensor: Drexel Biomedical Engineering laboratory; Light sources: Epitex Inc. Wavelength: 730, (805 not used) and 850 nm. 16 channels. Interoptode position: 2.5 cm. Sampling rate: 2 Hz. | OxyHb. | PFC. | Low-pass filter with a finite impulse response filter at 0.14 Hz to eliminate possible respiration, heart rate signals, and unwanted high-frequency noise. Using a combined independent component analysis/principal component analysis, environmental and equipment noise and signal drifts were removed from the raw intensity measurements. | Significant age × condition interaction, whereby significant increase in activation levels in PFC bilaterally in WWT compared with NW. The increase is also significantly more pronounced in young compared with the old. | Time course of trial was short as well as the time of baseline condition. | OxyHb levels are increased in the PFC during walking while talking compared with normal walking in healthy young and HOA. HOA may underutilize the PFC in attention-demanding locomotion tasks. |

**Table 1** (*continued*)

| Author, year | Sample | Aims | Gait assessment | Secondary task | Equipment details | Measurable parameters | Brain areas | Controlling, artefacts and filtering | Results | Limitations | Conclusion |
|---|---|---|---|---|---|---|---|---|---|---|---|
| *Holtzer et al. (2015)* | 348 HOA, 76.8 ± 6.8 years old, 59% female. | To determine the role that the PFC has in allocating attention resources to gait under single and DT conditions in HOA. | 3 continuous loops of walking on an electronic pathway (6 straight line trials of 4.5 m-long) at self-selected speed. Baseline: 10s standing trial pre-walk, counting silently in the head at rate of one number per second. Gait outcome: gait speed. Additional condition: standing for 30s while reciting alternate alphabet letters | Reciting alternate letters of the alphabet while walking 3 continuous loops as in the single walking task (6 straight line trials of 4.5 m-long). Instruction to pay equal attention to their walking and talking. Outcomes: letter rate and errors per minute. | Imager 1000 (fNIRS Devices LLC, Potomac, MD). Wavelength: 730, (805 not used) and 850 nm. 16 channels Interoptode position: 2.5 cm. Sampling rate: 2 Hz. | OxyHb. | PFC. | Lighting in the test room: 150lx or 1/3 of typical office lighting. Low-pass filter with a finite impulse response filter at 0.14 Hz to eliminate possible respiration, heart rate signals, and unwanted high-frequency noise. | OxyHb levels in all 16 channels were significantly higher in walking while talking trials compared to normal walking trials. Elevated PFC OxyHb levels were maintained throughout the course of walking while talking but not during the normal walking condition. Increased oxygenation levels in the PFC were related to greater stride length and better cognitive performance but not to faster gait velocity in walking while talking. Increased OxyHb levels during walking while talking were related to increased rate of correct letter generation. | | PFC plays a functional role in monitoring and allocating cognitive resources during locomotion, especially when cognitive demands are increased. |
| *Holtzer et al. (2017a)* | 318 HOA, 76.8 ± 6.7 years old, 178 women. | To determine the individual and combined effects of gender and perceived stress on the change on stride velocity and PFC OxyHb levels from normal walk to walk while talk conditions in HOA. | 3 continuous loops of walking on an electronic pathway (6 straight line trials of 4.5 m-long) at self-selected speed. Baseline: 10s standing trial pre-walk, counting silently in the head at rate of one number per second. Gait outcome: stride velocity. | Reciting alternate letters of the alphabet while walking 3 continuous loops as in the single walking task (6 straight line trials of 4.5 m-long). Instruction to pay equal attention to their walking and talking. Outcomes: letter rate and errors per minute. | Imager 1000 (fNIRS Devices LLC, Potomac MD). Wavelength: 730, (805 not used) and 850 nm. 16 channels. Interoptode position: 2.5 cm. Sampling rate: 2 Hz | OxyHb. | PFC. | To eliminate possible respiration, heart rate signals, and unwanted high-frequency noise, raw intensity measurements at 730 and 850 nm were low-pass filtered with a finite impulse response filter with a cut-off frequency of 0.14 Hz. Motion artefacts remained were eliminated by visual inspection of an expert fNIRS data analyst. Lighting in the test room: 150lx or 1/3 of typical office lighting. | Attenuation in increase in OxyHb levels, in high compared to low perceived stress levels, from the two single-task conditions to walking while talk was observed only in men. | | Older may be more vulnerable to the effect of perceived stress on the change in PFC OxyHb levels across walking conditions that vary in terms of cognitive demands. |
| *Holtzer et al. (2016)* | (i) 167 HOA with no gait impairments, 74.4 ± 6.0 years old, 85 women; (ii) 40 older adults with peripheral neurological gait abnormalities, 77.0 ± 6.3 years old, 17 women; (iii) 29 older adults with central neurological gait abnormalities, 79.6 ± 7.4 years old, 20 women. | To determine the effect of neurological gait abnormalities on the functional neural correlates of locomotion in older adults with regards to the posture first hypothesis. | 3 continuous loops of walking on an electronic pathway (6 straight line trials of 4.5 m-long) at self-selected speed. Baseline: 10s standing trial pre-walk, counting silently in the head at rate of one number per second. Gait outcome: stride velocity. | Reciting alternate letters of the alphabet while walking 3 continuous loops as in the single walking task (6 straight line trials of 4.5 m-long). Instruction to pay equal attention to their walking and talking. Outcomes: letter rate and errors per minute. | Imager 1000 (fNIRS Devices LLC, Potomac MD). Wavelength: 730, (805 not used) and 850 nm. 16 channels. Interoptode position: 2.5 cm. Sampling rate: 2 Hz | OxyHb. | PFC. | To eliminate possible respiration, heart rate signals, and unwanted high-frequency noise, raw intensity measurements at 730 and 850 nm were low-pass filtered with a finite impulse response filter with a cut-off frequency of 0.14 Hz. Noise (saturation or dark current conditions) was observed in 4% of the data that were subsequently excluded. | Higher oxyHb levels during walking while talking significantly higher compared with normal walk in normal. Central neural gait abnormalities was associated with significantly attenuated changes in oxyHb levels in walking while talking compared to single walking task. Among participants without neurological gait abnormalities, higher oxyHb levels (versus lower oxyHb levels-median split) were related to better cognitive performance, but slower gait velocity. In contrast, higher oxyHb levels during walking while talking among older adults with peripheral neurological gait abnormalities were associated with worse cognitive performance, but faster gait velocity. | Unknown if lighting conditions were controlled for. | Neural confirmation of the posture first hypothesis emerged among older adults whose postural and locomotive abilities were compromised. Increased activation in the PFC during locomotion may have a compensatory function that is designed to support gait among older adults with peripheral neurological gait abnormalities. |

Pelicioni et al. (2019), *PeerJ*, DOI 10.7717/peerj.6833

**Table 1** (*continued*)

| Author, year | Sample | Aims | Gait assessment | Secondary task | Equipment details | Measurable parameters | Brain areas | Controlling, artefacts and filtering | Results | Limitations | Conclusion |
|---|---|---|---|---|---|---|---|---|---|---|---|
| Holtzer et al., 2017b | 314 HOA, 76.8 ± 6.7 years old, 176 women. | To determine whether subjective fatigue was associated with objective fatigue measures, assessed during single and attention-demanding DT walking conditions, in HOA. | 3 continuous loops of walking on an electronic pathway (6 straight line trials of 4.5 m-long) at self-selected speed. Baseline: 10s standing trial pre-walk, counting silently in the head at rate of one number per second. Gait outcome: stride velocity. | Reciting alternate letters of the alphabet while walking 3 continuous loops as in the single walking task (6 straight line trials of 4.5 m-long). Instruction to pay equal attention to their walking and talking. Outcomes: letter rate and errors per minute. | Imager 1000 (fNIRS Devices LLC, Potomac MD). Wavelength: 730, (805 not used) and 850 nm. 16 channels. Interoptode position: 2.5 cm. Sampling rate: 2 Hz | OxyHb. | PFC. | To eliminate possible respiration, heart rate signals, and unwanted high-frequency noise, raw intensity measurements at 730 and 850 nm were low-pass filtered with a finite impulse response filter with a cut-off frequency of 0.14 Hz. Saturation or dark current conditions were excluded. | OxyHb levels significantly increased in walking while talking compared to normal walk. Worse subjective fatigue attenuated the increase in oxyHb from normal walking to walking while talking but it did not moderate changes in stride velocity. Worse subjective fatigue did not moderate changes in oxyHb during the course of the normal walk but was associated with attenuated oxyHb levels in the fourth, fifth and sixth straight walks compared to the first during walk-while-talk. | Unknown if lighting conditions were controlled for. | Worse subjective fatigue was associated with attenuation in both the increase of oxyHb levels from normal walking to walking while talking and the trajectory of oxyHb during the course of walking while talking. The trajectory of oxyHb during simple walking, however, was not associated with subjective fatigue. |
| *Koenraadt et al. (2014)* | 11 healthy young adults, 23 ± 4 years old, 8 women. | To understand the neural mechanism of precision stepping in comparison with normal gait. Also, to evaluate the role of different motor areas in the neural control of gait. | Walking on a treadmill at 3 km/h for 35s (10 trials). Baseline: 25 to 35s quiet standing prior to each walking trial. Gait outcome: step time variability. | Precision stepping walking on a treadmill at 3 km/h (10 trials). | Oxymon (Artinis Medical System, Zetten, the Netherlands) Wavelengths: 764 and 858 nm. 6-channel motor cortices unit and a 3-channel PFC unit. Interoptode position 1.0 and 4.0 cm. Sampling rate: 25 Hz. | OxyHb and deoxyHb. | Left hemisphere: S1, M1, SMA, pre-SMA and PFC. | Second order low-pass Butterworth filter with a cut-off frequency of 1.25 Hz was conducted to reduce high frequency noise. A second order high-pass Butterworth filter with a cut-off frequency of 0.01 Hz was used to reduce low frequency drift cause by fNIRS. After the correction for superficial interference, a second order low pass Butterworth filter with a cut-off frequency of 1 Hz was conducted. Continuous blood pressure monitoring. Short separation channels were used to remove hemodynamic changes in superficial tissue layers. | Task effect: Increased activation in the significant upper PFC channel (larger deoxyHb decrease) during the early-task (6–18.5s) for precision stepping compared to normal walking. No significant difference in motor cortices activation between the 2 walking conditions for either early or late phase. Phase effect in PFC: normal walking: increased activation in pre-task vs. early and late-task in normal walking (larger deoxyHb levels for early and late-task compared with pre-task) and versus late-task in precision stepping. Phase effect for motor cortices activation: in normal walking, larger pre-task oxyHb compared with early and late-task. The SMA, M1, and S1 revealed no significant differences between normal walking and precision stepping. | The authors pointed as limitation the small number of optodes used in this study. | The lack of M1/S1 activation during gait suggests that even in the current precision stepping task the control of ongoing gait depended mostly on subcortical automatisms, while motor cortices contributions did not differ between standing and walking. A prolonged activation of the PFC for precision stepping indicated that more attention was needed to perform precision stepping in comparison to normal walking. |

| Author, year | Sample | Aims | Gait assessment | Secondary task | Equipment details | Measurable parameters | Brain areas | Controlling, artefacts and filtering | Results | Limitations | Conclusion |
|---|---|---|---|---|---|---|---|---|---|---|---|
| *Lin & Lin (2016)* | 24 healthy young adults, 20–27 years old, 12 women. | To investigate the influence of cognitive task complexity and walking road condition on the neutral correlates of executive function and postural control in DT walking. | Walking on a 20 m walkway, 2 m in width for 60s (1 trial). Baseline: 40s quiet standing while fixating a cross on a smartphone held by participant (20s prior and 20s after the task). Gait outcomes: spatiotemporal and kinematic measures. | All tasks were performed for 60s (split in 3 20s periods to analyse time effects) -2 trials per condition: (i) walking on a narrow pathway (0.3 m width); (ii) obstacles (5 traffic cones) avoidance; (iii) easy working memory task (1-back task); (iv) hard working memory task (3-back task). Cognitive tasks outcomes: ratio of the number of correct responses to the number of all responses; average reaction time of the correct responses. No explicit instructions regarding task prioritization. | PortaLite fNIRS system (Artinis Medical Systems, the Netherlands). Wavelengths: 760 and 850 nm. 3 channels. Interoptode position 4.0 cm. Sampling rate: 50 Hz. | OxyHb. | PFC. | Data was low-pass filtered with a finite impulse response filter with a cut-off frequency of 0.2 Hz to attenuate the noises from non-evoked neurovascular coupling. | OxyHb levels changed significantly over time in all conditions. Relative changes in oxyHb concentration levels were all significantly different across the task complexity and walking conditions. OxyHb levels were all lower during DT walking than normal walking. Compared to wide and obstacle conditions, walking on the narrow road was found to elicit a smaller decrement in oxyHb levels. No significant correlation between the RT and the accuracy of the b-back tasks and fNIRS data. | No mention of whether movement artefacts were controlled for in the data processing. Unknown if lighting conditions were controlled for. | Healthy young adults are inclined to focus on the challenging working memory task and sacrificed gait performance to some extent through altered neural activations in the PFC and adapted coordination of lower-extremity kinematics. |
| *Lu et al. (2015)* | 17 healthy young adults, 23.1 ± 1.5 years old, 8 women. | (i) To evaluate the declines in gait performance caused by differing DT interference; (ii) to assess the alterations in cortical activation in the PFC, PMC and SMA when walking and performing a second cognitive or motor task compared with walking at a normal pace; (iii) To investigate the association between cortical activation and gait performance during DT. | Walking at self-selected pace over a 5.50 m long- electronic walkway for 60s (split in early phase: 5-20s and late phase: 21–50s) (3 trials). The first 5s and the last 10 were excluded due to hemodynamic effects. Baseline: 5s standing still. Gait outcomes: Temporal-spatial measures. | 3 trials in each condition, block-randomised: (i) walking while performing a cognitive task (subtracting 7 from an initial 3-digit number); (ii) walking while performing a motor task (carrying a bottle on a tray). Tasks duration and baseline condition are equal the gait assessment. | NIRSport (NIRx Medical Technologies LLC, NY, USA). Wavelengths: 760 and 850 nm. 14 channels. Interoptode position: 3.0 cm. Sampling rate: 7.81 Hz. | OxyHb, deoxyHb and index of haemoglobin differential (oxyHb–deOxyHb). | PFC, PMC and SMA | Data rejection based on coefficient of variation was used to reduce physical artefacts. The remaining fNIRS signals were bandpass-filtered (low-cut off frequency 0.01 Hz and high-cut off frequency 0.2 Hz) to eliminate the effects of heartbeat, respiration, and low-frequency signal drifts for each wavelength. Principal component analysis and spike rejection were used to correct for the motion artefacts. Some spikes were removed manually. | For the majority of channels, higher Hb differential during early and late phases of cognitive DT compared with simple walking. Stronger and more sustained brain activation, particularly in the PFC and PMC, during DT performances in cognitive DT compared with motor DT. Left PFC exhibited the strongest and most sustained activation during walking while performing a cognitive task compared with simple walking or motor DT. During DT activities, increased activation of the PMC and SMA were correlated with declines in gait performance. | The authors did not record the cognitive task and motor task performances in the DT conditions compared with the single tasks. | The negative relationship between PMC and SMA activation and gait variables suggests a control mechanism for maintaining gait performance during DT. |

*(continued on next page)*
**Table 1** (*continued*)

| Author, year | Sample | Aims | Gait assessment | Secondary task | Equipment details | Measurable parameters | Brain areas | Controlling, artefacts and filtering | Results | Limitations | Conclusion |
|---|---|---|---|---|---|---|---|---|---|---|---|
| *Lucas et al. (2018)* | 55 older adults, 74.8 ± 5.0 years old, 49% female. Participants were divided into a low white matter integrity group (*n* = 27) and a medium-high white matter integrity group (*n* = 28). | To examine the relationship between white matter microstructural integrity and changes in PFC oxyHb during active walking in older adults. | 3 continuous loops of walking on an electronic pathway (6 straight line trials of 4.5 m-long) at self-selected speed. Baseline: 10s standing trial pre-walk, counting silently in the head at rate of one number per second. Gait outcome: stride velocity. | Reciting alternate letters of the alphabet while walking 3 continuous loops as in the single walking task (6 straight line trials of 4.5 m-long). Instruction to pay equal attention to their walking and talking. Outcomes: letter rate and errors per minute. | Imager 1000 (fNIRS Devices LLC, Potomac MD). Wavelength: 730, (805 not used) and 850 nm. 16 channels. Interoptode position: 2.5 cm. Sampling rate: 2 Hz | OxyHb. | PFC. | To eliminate possible respiration, heart rate signals, and unwanted high-frequency noise, raw intensity measurements at 730 and 850 nm were low-pass filtered with a finite impulse response filter with a cut-off frequency of 0.14 Hz. Saturation or dark current conditions were excluded. Visual inspection to manually remove movement artefacts, saturation and dark current levels. | OxyHb levels increased from single to DT walking. White matter microstructural integrity moderated the effect of DT on PFC: after controlling for gait velocity, participants with deteriorated white matter integrity showed significantly greater increase in PFC oxyHB levels from single to DT walking, compared with participants with better white matter integrity. | No mention of controlling for external light. The authors pointed that the use of short source-detector channels and use of advanced filtering should be used in future studies. | Compromised white matter microstructural integrity may be a mechanism underlying inefficient brain response to increased cognitive demands of locomotion. |
| *Maidan et al. (2016)* | 68 people with PD (mild to moderate stages), 71.7 ± 1.1 years old, 22 women; 38 HOA, 70.4 ± 0.9 years old, 18 women. | (i) To examine changes in PFC activation during obstacle negotiation and DT walking, as compared with normal-walking in HOA; (ii) to investigate changes in PFC activation during normal-walking, DT and obstacle negotiation in people with PD; (iii) to compare PFC activation during the walking conditions between HOA and people with PD. | Overground walking at a self-selected pace (30m) for 30s (5 trials). Baseline: 5 s standing still (out of 20s before and 20s after each trial). Gait outcomes: gait speed and stride length. | 5 trials in each condition: (i) DT: walking while serially subtracting 3 from a given 3-digit number. The outcomes were: gait speed, stride length, DT cost and percentage of correct response; (ii) obstacle negotiation. The outcomes were: gait speed, duration of stepping over the obstacle and percentage change in step duration between steps over obstacle and normal steps. | PortaLite fNIRS system, (Artinis Medical Systems, Elst, the Netherlands). Wavelengths: 760 and 850 nm. 2 channels. Interoptode position 3.0, 3.5 and 4.0 cm amongst them. Sampling rate: 10 Hz. | OxyHb and deoxyHb. | PFC. | Probes were shielded from ambient light by covering the forehead with black fabric. A bandpass filter with frequencies of 0.01 to 0.14 Hz was used to reduce physiological noise (drift of the signal and heart beat). To remove motion artefacts, a wavelet filter was used. | People with PD had significant higher activation during normal walking compared with HOA. Significant group by condition interaction: during DT, oxyHb increased only in HOA. In contrast, oxyHb increased significantly during obstacle negotiation compared with simple walking, in people with PD. Significantly greater relative increase in oxyHb during DT compared with usual walking, for the HOA compared with the PD participants. In PD, higher PFC activation associated with faster gait speed. More clinical symptoms in PD were associated with lower PFC activation in simple walking and obstacle conditions. | Limited number of channels (2) used in this assessment. Also, the authors did not control the blood flow or heart rate even a filter was used for that proposal. The order of the trials was not randomized. | A different pattern of PFC activation during walking was observed between HOA and people with PD. The higher activation during normal walking in people with PD suggests that the PFC plays an important role already during simple walking. However, higher activation relative to baseline during obstacle negotiation and not during DT in people with PD shows that PFC activation depends on the nature of the task. |

Peer J

| Author, year | Sample | Aims | Gait assessment | Secondary task | Equipment details | Measurable parameters | Brain areas | Controlling, artefacts and filtering | Results | Limitations | Conclusion |
|---|---|---|---|---|---|---|---|---|---|---|---|
| *Maidan et al. (2018)* | 20 healthy young adults, 30.1 ± 1.0 years old, 10 women. | To explore the effects of obstacle height and available response time on PFC activation. | One trial of overground walking on an elliptical path of 50m. 4.5s before each obstacle. Gait outcomes: gait speed and stride length. | 3 trials in each of 4 conditions: anticipated and unanticipated obstacles with 2 different heights (50 and 100 mm). The baseline condition and the gait outcomes were the same as the simple walking. 3 phases of 3s analysed: before, over and after obstacles. | PortaLite fNIRS system, Artinis Medical Systems. Wavelengths: 760 and 850 nm. Sampling rate: 10 Hz. | OxyHb. | PFC. | Probes were shielded from ambient light by covering the forehead with black fabric. A bandpass filter with frequencies of 0.01 to 0.14 Hz was used to reduce physiological noise (drift of the signal and heart beat). To remove motion artefacts, a wavelet filter was used. | PFC activation significantly greater when stepping over all obstacles compared before and after obstacle crossing. PFC activation during obstacle negotiation is not affected by obstacle height Significant effect of available response time: during unanticipated obstacles, the slope of the oxyHb response was steeper, as compared to anticipated obstacles. These changes in PFC activation during negotiation of unanticipated obstacles were correlated with greater distance of the leading foot after the obstacles. | The authors did not report on channel numbers and interoptodes distance. The authors pointed as a limitation that the use of harness might have affected the participants' walking. Regarding the unanticipated obstacle the participants could estimate the location of the same obstacle in other trials. | The pattern of PFC activation depends on the nature of obstacle. During unanticipated obstacles the recruitment of PFC is faster and greater than during negotiation of anticipated obstacles. |
| *Meester et al. (2014)* | 17 healthy young adults, 27.8 ± 6.3 years old, 10 women. | To assess the effect of speed and cognitive load on the automatic processing (PFC activity, spinal cord activity and gait) during walking in healthy young adults. | Walking on a treadmill at a self-selected pace and 20% faster for 30s (5 trials at normal speed, fast speed, with and without secondary task). Baseline: 10s standing still in the middle of resting time between trials. Gait outcome: step time. | Cognitive conditions: counting back by seven; 10s in the middle of the trials used for statistical analyses. | Oxymon (Artinis Medical System, The Netherlands). Wavelengths: 782 and 859 nm. 4 channels. Interoptode position 3.0 cm. Sampling rate: 10 Hz. | OxyHb and deoxyHb. | PFC. | Blood pressure and heart rate were measured at the start and end of testing session. OxyHb and deoxyHb were calculated and filtered with a low pass filter set at 0.67 Hz and visually inspected for motion artefacts, missing signals and noisy signals. Blocks with missing signals or artefacts were excluded. | OxyHb concentrations significantly increased in the right PFC in the DT compared to a single task walking. PFC activity was unaffected by increases of walking speed. No significant correlation between PFC activation, H-reflex variability and step time variability. | The authors pointed the use of treadmill as a limitation (no ecological validity).No mention of controlling for external light. The authors point the high variability of the fNIRS responses between individuals. | Healthy young adults increased PFC activity in response to increasing cognitive load but maintained gait performance and reflex activity. The increases in PFC activity allowed individuals to perform additional tasks simultaneously without affecting cortical output onto the measured peripheral reflexes and thus gait control. |
| *Mirelman et al. (2014)* | 23 healthy young adults, 30.9 ± 3.7 years old, 13 women. | To investigate whether an increase in frontal activation is specific to DT during walking. | 5 trials of walking on a 30 m walkway at self-selected speed. Baseline: 20s standing still. Gait outcomes: spatiotemporal variables. | (i) walking while counting forward; (ii) walking while serially subtracting 7 from a pre-determined 3 digit number; (iii) standing in place while serially subtracting 7 from a pre-determined 3 digit number (Walking + S7). Tasks duration and baseline condition are equal to the gait assessment. | Artinis, The Netherlands. Wavelengths: 760 and 850 nm. 6 channels. Sampling rate: 10 Hz. | OxyHb. | PFC. | To eliminate physiologically irrelevant effects, a low-pass filter was applied with a finite impulse response filter, with a cut-off frequency at 0.14 Hz before processing the signals. | Walking alone demonstrated the lowest levels of oxyHb followed by walking + counting condition, followed by Walking + S7 condition, which was significantly different compared to the two other walking conditions. No significant differences in oxyHb levels were observed between usual walking and the standing condition or between standing with or without serial subtraction. During walking + S7, gait variability and the number of subtractions completed were inversely correlated with oxyHb levels. | The order of the tasks was not randomised. No mention of controlling for ambient light artefacts. | DT during walking is associated with frontal brain activation in healthy young adults. The observed changes are apparently not a response to the verbalization of words and are related to the cognitive load during gait. |

**Table 1 (*continued*)**

| Author, year | Sample | Aims | Gait assessment | Secondary task | Equipment details | Measurable parameters | Brain areas | Controlling, artefacts and filtering | Results | Limitations | Conclusion |
|---|---|---|---|---|---|---|---|---|---|---|---|
| *Mirelman et al. (2017)* | 23 healthy young adults, 30.9 ± 3.7 years old, 57% female; 20 HOA, 69.7 ± 5.8 years old, 50% female. | To study the effects of aging on gait and PFC activation in complex walking task with internal and external task demands. | 5 trials of walking on a 30 m walkway at self-selected speed. Baseline: 20s standing still pretrial. Gait outcomes: spatiotemporal variables from an electronic mat placed in the middle of the walkway. | (i) walking while serially subtracting 7 from a predetermined 3 digit number; (ii) walking while negotiating two physical obstacles. Tasks duration and baseline condition are equal to the gait assessment. The DT score was also calculated. | Portalite Artinis, The Netherlands. Wavelengths: 760 and 850 nm. 6 channels. Interoptode distances were: 30, 35 and 40 mm. Sampling rate: 10 Hz. | OxyHb. | PFC. | A band-pass filter with frequency of 0.01–0.14 Hz was used to reduce physiological noise such as heart beat and drift of the signal. To remove motion artefacts a wavelet filter was used, followed by correlation based signal improvement. Probes were shielded from ambient light using a black cloth. | HOA had significant increases in oxyHb levels during simple walking, relative to standing. Both groups showed significantly increased PFC activation in the DT and in the obstacle negotiation conditions compared with usual walking. Significant group effect whereby younger participants had lower oxyHb levels than HOA, in all conditions. Significant positive correlation between age and oxyHb levels, as well as between oxyHb levels and gait variability in HOA in the obstacle condition. Age was a significant independent predictor of oxyHb levels in the usual walking. Age and gait speed were significant independent predictors of oxyHb levels in the obstacle condition but not the DT. | The authors pointed the limitations as: small sample size; few probes to analyse the brain activation in the PFC and also, that, other brain areas (motor cortex) should be investigated in future studies. | PFC activation during walking is dependent on age and task properties. HOA apparently rely more on cognitive resources even during usual walking. |
| *Mori, Takeuchi & Izumi (2018)* | 14 stroke survivors, 61.1 ± 9.3 years old, 2 women; 14 healthy subjects, 66.3 ± 13.3 years old, 3 women. | To investigate the association between PFC activity and DT interference on physical and cognitive performance in stroke survivors. | Walking at a comfortable pace around a circle with a radius of 2.5 m for 60s. Baseline: 40s prior to calculation task and 20s after the task. The outcome measured was the trunk linear acceleration. | 3 blocks (i) control period: the participants were instructed to repeat the number 1–10 in sequence; (ii) calculation period: participants performed subtractions of 3, beginning with a random number between 100 and 199. Participants performed calculation tasks while standing and walking. Mean values of correct and mistaken answers in each condition were compared. | WOT (Hitachi Corporation, Japan) Wavelengths: 705 and 830 nm. 16 channels. Interoptode position 3.0 cm. Sampling rate: 5 Hz. | OxyHb. | PFC. | A band pass filter with a low pass (0.5 Hz) was applied to account for the effects of Mayer waves and high-frequency fluctuations, whereas that with a high pass (0.01 Hz) was used for baseline drift. | PFC activation during DT walking was significantly lower in stroke survivors. Right PFC activation was negatively correlated with DT cost on acceleration magnitude in stroke survivors, yet not in healthy participants. In healthy participants, left PFC activation was significantly negatively correlated with correct rate and mistake rate of subtractions. No significant correlation between PFC activation and cognitive performance | The authors did not apparently control for light interference. The authors did not measure objectively some physiological responses (i.e. blood pressure, heart rate). | During DT walking, PFC activation might prioritize physical demands in stroke survivors, but might prioritize cognitive demands in healthy subjects. The results suggest that during DT walking, the stroke patients prioritize their motor demands while healthy subjects prioritize the cognitive task. |

Pelicioni et al. (2019), *PeerJ*, DOI 10.7717/peerj.6833

**Table 1** (*continued*)

| Author, year | Sample | Aims | Gait assessment | Secondary task | Equipment details | Measurable parameters | Brain areas | Controlling, artefacts and filtering | Results | Limitations | Conclusion |
|---|---|---|---|---|---|---|---|---|---|---|---|
| *Nieuwhof et al. (2016)* | 12 people with PD (mild to moderate stages), 70.1 ± 5.4 years old, 5 women. | (i) To examine the feasibility of measuring bilateral PFC activity in people with PD during different DT walking conditions using two wireless fNIRS devices; (ii) To investigate whether it is possible to record the expected typical fNIRS signal of neuronal activity in the PFC as consequence of DT walking compared to rest; (iii) To investigate the sensitivity of the method to detect oxyHb and deoxyHb concentrations between DT walking and simple walking. | 5 trials of walking back and forth over a course of approximately 8 m at a self-selected pace for 40s. Baseline: 20s standing still before and after trials (final 5s of these 20s periods were used in the analysis). Gait outcomes: temporal-spatial. | Five blocks each including each of the three cognitive tasks conditions: (i) walking while counting forward; (ii) walking while serially subtracting 7 from a pre-determined 3 digit number; (iii) reciting digit spans. Outcomes: the number of subtractions and digit spans completed within the task and the percentage of correct answers on both tasks. | PortaLite fNIRS system, Artinis Medical Systems. Wavelengths: 760 and 850 nm. Interoptode position 3.0, 3.5 and 4.0 cm amongst them. Sampling rate: 10 Hz. | OxyHb and deoxyHb. | PFC. | The devices were shielded from ambient light by covering the whole forehead with a black cloth. The movement artefact reduction algorithm was performed within each trial. The fNIRS signals were also linearly de-trended per trial and low-pass filtered at 0.1 Hz using a Butterworth filter to remove heart rate and other higher frequency physiological signals. Before starting any trial, participants stood for at least 1 min to minimize blood pressure fluctuations after standing up. | Bilateral PFC oxy-Hb concentrations were significantly increased during walking while serially subtracting and reciting digit spans when compared to rest. DeoxyHb concentrations did not differ between the walking tests and rest. | The authors pointed as limitations: the small sample; that they did not control blood pressure simultaneously with fNIRS that they did not control for superficial hemodynamics using short reference channels for example. | Using the new wireless fNIRS devices described in this paper, it is feasible to measure the PFC activity in PD during DT walking. |
| *Osofundiya et al. (2016)* | 20 community-dwelling older adults: (i) 10 non-obese, 80.6 ± 7.5 years old, 8women; (ii) 10 obese, 80.5 ± 6.8 years old, 6 women. | To determine the obesity-specific activation of the PFC using fNIRS during simple and complex ambulatory tasks in older adults. | Walking back and forth at a self-selected pace for 30s (4 trials). Baseline: 30s quiet seating. 10s quiet standing in between each trial and 2 min of seated rest between blocks. Gait outcome: gait speed. | 2 blocks of simple and complex walking trials. Each block includes 4 30-s long trials (i) walking + cognitive DT (recite alternate letters of the alphabet) (outcome: percent correct responses); (ii) precision walking (stepping on surface targets) (outcome: average number of targets attained). Tasks duration and baseline condition are equal to the gait assessment Participants encouraged to perform their best on the secondary tasks. | NIRO 200 NX, (Hamamatsu Photonics, Japan). 2 channels. Sampling rate: 5 Hz. | OxyHb and total Hb levels. | PFC. | Heart rate was continuously monitored using a heart rate monitor and was averaged across each trial. Participants instructed to avoid any sudden head movements during the tasks. Probes covered with a black headband to eliminate external lights. | Significant task * group interactions on oxyHb levels. Significantly increased oxyHb levels in DT and precision walking tasks compared to rest and usual walking, as well as in obese compared with non-obese individuals. Obesity was associated with three times greater oxyHb levels, particularly during the precision gait task, despite obese adults demonstrating similar gait speeds and performances on the complex gait tasks as non-obese adults. | The authors pointed: small sample size might limit the extension of these results; some conditions were not controlled (diabetes, higher blood pressure, etc.). Order of simple and DT trials counterbalanced but precision walking always last. | In order to maintain gait performance, obesity was associated with higher neural costs, and this was augmented during ambulatory tasks requiring greater precision control. |

**Table 1** (*continued*)

| Author, year | Sample | Aims | Gait assessment | Secondary task | Equipment details | Measurable parameters | Brain areas | Controlling, artefacts and filtering | Results | Limitations | Conclusion |
|---|---|---|---|---|---|---|---|---|---|---|---|
| *Suzuki et al. (2004)* | 9 healthy young adults, 28.1 ± 7.4 years old, 2. | To assess cortical activation patterns associated with locomotor speed as assessed by relative changes of oxyHb and deoxyHb levels using fNIRS. | Three locomotor tasks: walking at 3 km/h, 5 km/h and walking at 9 km/s on a treadmill for 90s (1 trial). Baseline: 20s standing still. Gait outcome: cadence. 60s rest between trials (30s before, 30s after). | Increase in locomotor speed. Task data in the 13-s period just before reaching each constant speed were used for analysis. | OMM-2001(Shimadzu, Japan). Wavelengths: 780, 805 and 830 nm. 42 channels. Interoptode position 3.0 cm. Sampling rate: 5.26 Hz. | OxyHb, deoxyHb and total Hb levels. | PFC, PMC, medial-SMC and lateral-SMC. | Blood pressure, heart rate and arterial oxygen saturation were measured immediately before and after each task | PFC activation was significantly greater when the participants ran at 9 km/h than when they walked at 3 km/h and at 5 km/h. Activations in the PFC, PMC and medial-SMC were significantly greater than that in the lateral SMC. | No mention of filtering or controlling for movement artefacts and ambient light in this study. | The PFC was significantly more activated during the periods before reaching a constant speed in the 9 km/h run compared with the 5 km/h walk and compared with the 3 km/h walk. PFC might be involved together with other structures in controlling locomotion to adapt to the increasing speed in the acceleration of phase of locomotion. |
| *Suzuki et al. (2008)* | 7 healthy young adults, 31.3 + 4.8 years old, 3 women. | To assess how a verbal instruction before walking would affect cortical activation and walking performance using fNIRS. | Treadmill walking at 3 km/h in 2 conditions: (i) Simple walking: 40 s walking; (ii) Prepared walking: after verbal instruction walking for 30s (10s standing pre-walking but post "ready" instruction, also recorded). 4 trials in each condition. Pseudo-randomized rest (10, 15, 20 and 25s). Baseline: 10s standing still. Gait outcomes: cadence and step length. | None. | OMM-2001 (Shimadzu, Japan). Wavelengths: 780, 805 and 830 nm. 42 channels. Interoptode position 3.0 cm. Sampling rate: 5.26 Hz. | OxyHb, deoxyHb and total Hb levels. | PFC, SMA, PMC, medial SMC and lateral SMC. | None. | Significant main effect for site of region whereby activations of the PFC, PMC, SMA and medial SMC were significantly enhanced by the preparation for walking during the rest and walking periods as indicated by large effect sizes in OxyHb levels. However, there were no significant changes in deoxyHb. Concurrent significant changes in walking performance whereby cadence was smaller and stride length larger in the prepared walking compared with the simple walking conditions. | No mention of filtering or controlling for movements, or controlling for ambient light. | Preparation of gait shares similar structures in the frontal cortex with gait execution. Preparation for gait enhanced frontal activation and influenced gait performance. |

**Table 1** (*continued*)

| Author, year | Sample | Aims | Gait assessment | Secondary task | Equipment details | Measurable parameters | Brain areas | Controlling, artefacts and filtering | Results | Limitations | Conclusion |
|---|---|---|---|---|---|---|---|---|---|---|---|
| *Takeuchi et al. (2016)* | 15 HOA, 71.7 ± 3.3 years old, 5 women; 15 healthy young adults, 25.9 ± 4.4 years old, 5 women. | To evaluate the correlation between PFC activity and DT cost during smartphone use while walking in young and HOA. | Baseline: overground walking on a 2.5 m radius pathway at self-selected speed for 30s (5 trials). Gait outcomes: step time and acceleration magnitude. | 5 trials in DT condition: performing a number-selecting task on a smartphone while walking for 30s. Cognitive-only task: same smartphone while they were seated. Cognitive outcomes: number of correct responses and errors. Tasks duration was similar across conditions. DT cost of cognitive and gait performance compute. Participants were instructed not to consciously prioritize either task. | WOT (Hitachi Corporation, Japan). Wavelengths: 705 and 830 nm. 16 channels. Interoptode position 3.0 cm. Sampling rate 5 Hz. | OxyHb | PFC, divided in left, middle and right PFC. | Rapid changes in oxyHb concentration more than 3 SD over the average for two consecutive samples were considered as movement artefacts. All blocks that were affected by these motion artefacts were excluded. A band pass filter of low pass 0.5 Hz was applied for the effects of Mayer waves and high-frequency fluctuations. A high-pass 0.01 Hz was used to account for baseline drift. Participants were instructed to keep their faces turned to the screen of the smartphone and to minimize head movements in all conditions. | No significant effect of age or site on oxyHb levels in the DT condition. In healthy young adults: significant positive associations between right PFC activation and DT cost on acceleration magnitude and negative association between left PFC activation and DT cost on error rate. In HOA: negative associations between middle PFC activation and DT cost on step time and between left PFC activation and DT on acceleration magnitude. DT costs on correct and mistake rates in the HOA group were significantly higher than in the young group. | The instruction to focus on the smartphone might have induced a cognitive task—first strategy. Some HOA are not used to using a smartphone properly. The authors raise the issue of recording skin blood flow, blood pressure and heart rate, as well as the possibility that cortical atrophy (scalp to cortex distance) might affect hemodynamic responses. | In healthy young adults, the left PFC inhibited inappropriate action and the right PFC stabilized walking performance during DT. PFC activity in HOA was less lateralized for supressing DT cost on gait performance during DT walking, resulting in inability to cope with a cognitive demand. |
| *Thumm et al. (2018)* | 20 people with PD, 69.8 ± 6.5 years old, 10 women. | To investigate whether during treadmill walking, PFC activation in people with PD is lower as compared to overground walking. | Five 30s-trials at self-selected speed in each condition in the following fixed order: (i) overground walk: in a 30 m corridor; (ii) treadmill walk. Baseline: 20s standing still before and after each-trial. Gait outcomes: gait speed and stride time. | None. | PortaLite fNIRS system, (Artinis Medical Systems, Elst, the Netherlands). Wavelengths: 760 and 850 nm. 2 channels. Interoptode position 3.0, 3.5 and 4.0 cm amongst them. Sampling rate: 10 Hz | OxyHb. | PFC. | The authors cited the methodology used in *Maidan et al. (2016)*: Probes were shielded from ambient light by covering the forehead with black fabric. A bandpass filter with frequencies of 0.01 to 0.14 Hz was used to reduce physiological noise (drift of the signal and heart beat). To remove motion artefacts, a wavelet filter was used. | OxyHb levels and gait speed were significantly lower during treadmill walking compared with overground walking. Gait stability was significantly enhanced in the anterior-posterior and the medial-lateral directions during treadmill walking compared with overground walking. According to regression analyses, age and disease duration were significantly associated with differences in oxyHb between conditions, while gait speed was not. | The authors reported that they did not control for superficial hemodynamics or systemic changes like blood flow and or heart rate or lower limb kinematics in the two walking conditions. | The findings support the idea that when gait is externally paced, PFC activation is reduced in people with PD, perhaps reflecting a reduced need for compensatory cognitive mechanisms. |

**Table 1** (*continued*)

| Author, year | Sample | Aims | Gait assessment | Secondary task | Equipment details | Measurable parameters | Brain areas | Controlling, artefacts and filtering | Results | Limitations | Conclusion |
|---|---|---|---|---|---|---|---|---|---|---|---|
| *Verghese et al. (2017)* | 166 HOA, 75.0 ± 6.1 years old, 85% women. | To determine whether PFC activity during walking predicts falls in HOA. | 3 continuous loops of walking on an electronic pathway (6 straight line trials of 4.5 m-long) at self-selected speed. Baseline: 10s standing trial pre-walk, counting silently in the head at rate of one number per second. Gait outcome: stride velocity. | Reciting alternate letters of the alphabet while walking 3 continuous loops as in the single walking task (6 straight line trials of 4.5 m-long). Instruction to pay equal attention to their walking and talking. Outcomes: letter rate and errors per minute. | Imager 1000, fNIRS Devices LLC. Wavelength: 730 and 850 nm. 16 channels Interoptode position: 2.5 cm. Sampling rate: 2 Hz | OxyHb. | PFC. | Raw intensity measurements at 730 and 850 nm were low-pass filtered with a finite impulse response filter with a cut-off frequency of 0.14 Hz. This procedure was adopted to eliminate artefacts due to respiration, heart rate signals and unwanted high frequency noise. | Higher PFC activation levels on fNIRS during walking while reciting alphabet letters predicted falls over a 50-month follow-up. Each 1SD increase in the intensity of PFC during the DT was associated with a 32% increased risk of falls. Only activation in the left PFC predicted falls. | Unknown if lighting conditions were controlled for | PFC activity levels while performing a cognitively demanding walking condition predicted falls in HOA. |

**Notes.**

PFC, Prefrontal cortex; DT, dual-task; OxyHb, oxygenated hemoglobin; DeoxyHb, deoxygenated hemoglobin; fNIRS, functional near-infrared spectroscopy; TOI, tissue oxygenated index; MS, Multiple Sclerosis; HOA, healthy older adults; MCI, mild cognitive impairment; PMC, premotor cortex; SMA, supplementary motor area; SMC, sensorimotor cortex; S1, primary sensorimotor cortex; M1, primary motor cortex; Hb, hemoglobin; BA, Brodmann area; PD, Parkinson's disease.

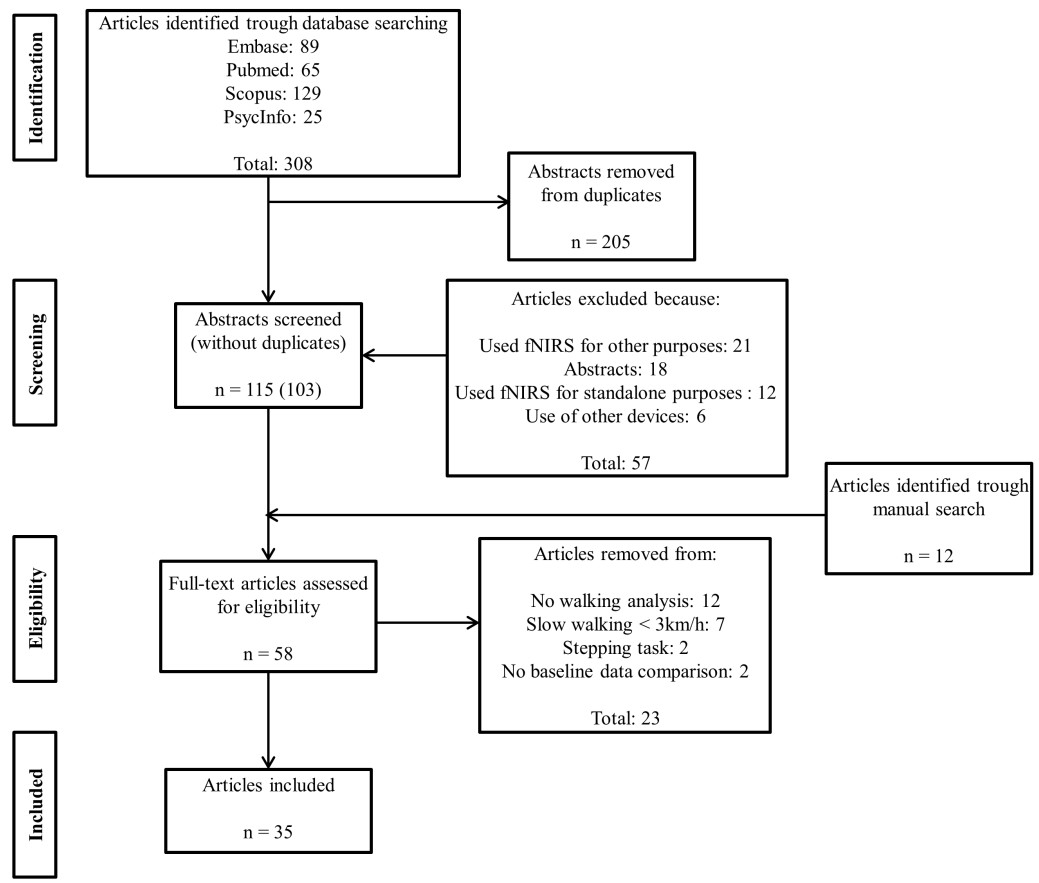

**Figure 1** **Flowchart of the article selection process.**

and clinical groups with balance disorders (simple walking (1), motor and cognitive tasks (2), somatosensory and cognitive tasks (1), only cognitive tasks (4)); 3 studies involved solely clinical groups with balance disorders (simple walking (1) and only cognitive tasks (2)). These studies are summarized in Table 2.

Thirteen studies (*Beurskens et al., 2014*; *Chen et al., 2017*; *Clark et al., 2014a*; *Clark et al., 2014b*; *Hawkins et al., 2018*; *Lin & Lin, 2016*; *Lu et al., 2015*; *Maidan et al., 2016*; *Mirelman et al., 2014*; *Mirelman et al., 2017*; *Nieuwhof et al., 2016*; *Osofundiya et al., 2016*; *Maidan et al., 2018*) reported two or more comparisons, i.e., walking while crossing an obstacle and walking while performing a serial subtraction, both in comparison to a baseline condition such as simple walking. Many studies also contrasted brain activation in more than one area. In total, 75 brain activation comparisons are included in this review.

Table 3 presents the individual and overall methodological reporting scores, and also describes how the scores were attributed for each paper. Twenty studies (57%) were classified as high quality, 10 (29%) as medium quality and five (14%) as low quality. Figure 2 shows the number of comparisons from the included studies showing an increase, decrease or no change in PFC activation when comparing either (a) walking to standing or (b) walking with an additional task to simple walking, according to group. In young people,

**Table 2** Effect of the additional tasks relative to baseline conditions on regional cortical activation in healthy young adults, healthy older adults and clinical groups with balance disorders.

| | Healthy young adults | Healthy older adults | Clinical groups with balance disorders |
|---|---|---|---|
| *Simple walking* | | | |
| *Caliandro et al. (2015)[a]* | | PFC (=) | PFC (+)[k] |
| *Lin & Lin (2016)[a]* | PFC (+) | | |
| *Hawkins et al. (2018)[a]* | PFC (−) | PFC (+)[d], PFC (=)[e] | PFC (+)[d,l], PFC (=)[e,l] |
| *Thumm et al. (2018)[a]* | | | PFC (+)[m] |
| *Fast walking* | | | |
| *Harada et al. (2009)[a]* | | left PFC (+), right PFC (=), preSMA (=), SMA (+), mSMC (=) | |
| *Eggenberger et al. (2016)[b]* | | PFC (=) | |
| *Suzuki et al. (2004)[b]* | PFC (+), PMC (=), SMC (=) | | |
| *Motor task* | | | |
| *Chen et al. (2017)[b]* | | PFC (+)[f] | |
| *Clark et al. (2014b)[b]* | | PFC (+)[f] | |
| *Koenraadt et al. (2014)[b]* | PFC (−)[d], PFC (=)[e], preSMA (=), SMA (=), S1 (=), M1 (=) | | |
| *Lin & Lin (2016)[a]* | PFC (−) | | |
| *Lu et al. (2015)[b]* | left PFC (+), SMA (+) | | |
| *Maidan et al. (2016)[b]* | | PFC (=) | PFC (+)[m] |
| *Maidan et al. (2018)[b]* | PFC (+) | | |
| *Mirelman et al. (2017)[b]* | PFC (+) | PFC (+) | |
| *Osofundiya et al. (2016)[a,b]* | | PFC (+) | PFC (+)[n] |
| *Hawkins et al. (2018)[a]* | PFC (−) | PFC (=) | PFC (+)[l] |
| *Somatosensory task* | | | |
| *Chaparro et al. (2017)[a,c]* | | PFC (+) | PFC (+)[o] |
| *Clark et al. (2014a)[b]* | | PFC (−) | |
| *Clark et al. (2014b)[b]* | | PFC (=) | |
| *Cognitive task* | | | |
| *Al-Yahya et al. (2016)[b]* | | PFC (=) | PFC (+)[l] |
| *Beurskens et al. (2014)[b]* | PFC (=) | PFC (−)[g], PFC (=)[h] | |
| *Chaparro et al. (2017)[a,c]* | | PFC (+) | PFC (+)[o] |
| *Chen et al. (2017)[b]* | | PFC (+) | |
| *Clark et al. (2014a)[b]* | | PFC (+)[i] | |
| *Clark et al. (2014b)[b]* | | PFC (+) | |
| *Doi et al. (2013)[b]* | | | PFC (+)[p] |
| *Hernandez et al. (2016)[b]* | | PFC (+) | PFC (+)[o] |
| *Hill et al. (2013)[c]* | left PFC (+) | | |
| *Hawkins et al. (2018)[a]* | PFC (=) | PFC (+)[d], PFC (=)[e] | PFC (+)[l] |
| *Holtzer et al. (2011)[b]* | PFC (+) | PFC (+) | |
**Table 2 (*continued*)**

| | Healthy young adults | Healthy older adults | Clinical groups with balance disorders |
|---|---|---|---|
| *Holtzer et al. (2015)[b]* | | PFC (+) | |
| *Holtzer et al. (2016)[b]* | | PFC (+) | PFC (+)[q] |
| *Holtzer et al. (2017a)[b]* | | PFC (+) | |
| *Holtzer et al. (2017b)[b]* | | PFC (+) | |
| *Lin & Lin (2016)[a]* | PFC (−) | | |
| *Lu et al. (2015)[b]* | PFC (+), PMC (+), SMA (+) | | |
| *Lucas et al. (2018)[b]* | | PFC (+) | |
| *Maidan et al. (2016)[b]* | | PFC (+) | PFC (=)[m] |
| *Meester et al. (2014)[b]* | right PFC (+) | | |
| *Mirelman et al. (2014)[b]* | PFC (+)[j] | | |
| *Mirelman et al. (2017)[b]* | PFC (+) | PFC (+) | |
| *Mori, Takeuchi & Izumi (2018)[a,b]* | | PFC (=) | PFC (=)[l] |
| *Nieuwhof et al. (2016)[a]* | | | PFC (+)[m] |
| *Osofundiya et al. (2016)[a,b]* | | PFC (+) | PFC (+)[n] |
| *Takeuchi et al. (2016)[a]* | PFC (=) | PFC (=) | |
| *Verghese et al. (2017)[a]* | | PFC (+) | |

**Notes.**

PFC, Prefrontal cortex; preSMA, Pre-supplementary motor area; SMA, Supplementary motor area; S1, Primary sensorimotor cortex; M1, Primary motor cortex; PMC, Premotor cortex; mSMC, Medial sensorimotor cortex; (+), higher activation when performing the additional task; (−), lower activation when performing the additional task; (=), no changes in activation.

Baseline comparison:
[a]Standing still.
[b]Simple walking.
[c]Easier level of the secondary task.
Significant differences only for:
[d]First half of the task.
[e]Second half of the task.
[f]Obstacle negotiation and wearing a vest with 10% body weight conditions.
[g]Walk vs. visual task.
[h]Walk vs alphabet recall.
[i]Overground walking.
[j]Walking and counting backwards task.
Clinical groups:
[k]Ataxia.
[l]Stroke.
[m]Parkinson's disease.
[n]Obesity.
[o]Multiple Sclerosis.
[p]Mild Cognitive Impairment.
[q]Neurological gait.

11 of the 20 comparisons from 13 studies reported significant increases in PFC activation (*Lu et al., 2015*; *Suzuki et al., 2004*; *Lin & Lin, 2016*; *Hill et al., 2013*; *Holtzer et al., 2011*; *Maidan et al., 2018*; *Mirelman et al., 2017*; *Meester et al., 2014*; *Mirelman et al., 2014*); five comparisons reported a reduction in PFC activation (*Hawkins et al., 2018*; *Koenraadt et al., 2014*; *Lin & Lin, 2016*) and four comparisons reported no change (*Koenraadt et al., 2014*; *Beurskens et al., 2014*; *Hawkins et al., 2018*; *Takeuchi et al., 2016*). Although the aim of the study was to report changes in PFC activation, three studies (four comparisons) have also investigated activation in additional cortical areas in young people (Table 1) (*Lu et al., 2015*; *Koenraadt et al., 2014*; *Suzuki et al., 2004*). Of these, two comparisons indicated

Pelicioni et al. (2019), *PeerJ*, DOI 10.7717/peerj.6833

**Table 3** Methodological reporting criteria ratings for the included studies.

| Study | Equipment details provided | Movement artefacts considered | Optode placement specified | External light confounding effect considered | Heart changes confounding effect considered | Sample size ($n > 10$ per group) | Score | Quality criteria |
|---|---|---|---|---|---|---|---|---|
| Al-Yahya et al. (2016) | 1 | 1 | 1 | 0 | 1 | 1 | 5 | High quality |
| Beurskens et al. (2014) | 1 | 1 | 1 | 1 | 1 | 1 | 6 | High quality |
| Caliandro et al. (2015) | 1 | 1 | 1 | 0 | 1 | 1 | 5 | High quality |
| Chaparro et al. (2017) | 1 | 0 | 0 | 0 | 1 | 1 | 3 | Medium quality |
| Chen et al. (2017) | 1 | 1 | 1 | 0 | 1 | 1 | 5 | High quality |
| Clark et al. (2014a) | 0 | 0 | 0 | 0 | 0 | 1 | 1 | Low quality |
| Clark et al. (2014b) | 0 | 0 | 0 | 0 | 0 | 1 | 1 | Low quality |
| Doi et al. (2013) | 1 | 1 | 1 | 0 | 0 | 1 | 4 | Medium quality |
| Eggenberger et al. (2016) | 1 | 1 | 1 | 0 | 1 | 1 | 5 | High quality |
| Harada et al. (2009) | 1 | 0 | 1 | 0 | 1 | 1 | 5 | High quality |
| Hawkins et al. (2018) | 1 | 1 | 0 | 0 | 0 | 0 | 2 | Low quality |
| Hernandez et al. (2016) | 1 | 1 | 1 | 1 | 1 | 0 | 5 | High quality |
| Hill et al. (2013) | 1 | 1 | 0 | 0 | 1 | 0 | 3 | Medium quality |
| Holtzer et al. (2011) | 1 | 1 | 1 | 0 | 1 | 1 | 5 | High quality |
| Holtzer et al. (2015) | 1 | 1 | 1 | 1 | 1 | 1 | 6 | High quality |
| Holtzer et al. (2016) | 1 | 1 | 1 | 0 | 1 | 1 | 5 | High quality |
| Holtzer et al. (2017a) | 1 | 1 | 0 | 0 | 1 | 1 | 4 | Medium quality |
| Holtzer et al. (2017b) | 1 | 1 | 0 | 0 | 1 | 1 | 4 | Medium quality |
| Koenraadt et al. (2014) | 1 | 1 | 1 | 0 | 1 | 1 | 5 | High quality |
| Lin & Lin (2016) | 1 | 0 | 1 | 0 | 0 | 1 | 3 | Medium quality |
| Lu et al. (2015) | 1 | 1 | 1 | 0 | 1 | 1 | 5 | High quality |
| Lucas et al. (2018) | 1 | 1 | 1 | 0 | 1 | 1 | 5 | High quality |
| Maidan et al. (2016) | 1 | 1 | 0 | 1 | 1 | 1 | 5 | High quality |
| Maidan et al. (2018) | 1 | 1 | 1 | 1 | 1 | 1 | 6 | High quality |
| Meester et al. (2014) | 1 | 1 | 1 | 0 | 1 | 1 | 5 | High quality |
| Mirelman et al. (2014) | 1 | 0 | 1 | 0 | 0 | 1 | 3 | Medium quality |
| Mirelman et al. (2017) | 1 | 1 | 1 | 1 | 1 | 1 | 6 | High quality |
| Mori, Takeuchi & Izumi (2018) | 1 | 1 | 1 | 0 | 1 | 1 | 5 | High quality |
| Nieuwhof et al. (2016) | 1 | 1 | 0 | 1 | 1 | 1 | 5 | High quality |
| Osofundiya et al. (2016) | 0 | 0 | 1 | 1 | 1 | 1 | 4 | Medium quality |
| Suzuki et al. (2008) | 1 | 0 | 0 | 0 | 0 | 0 | 1 | Low quality |
| Suzuki et al. (2004) | 1 | 0 | 0 | 0 | 1 | 0 | 2 | Low quality |

Pelicioni et al. (2019), *PeerJ*, DOI 10.7717/peerj.6833

**Table 3** (*continued*)

| Study | Equipment details provided | Movement artefacts considered | Optode placement specified | External light confounding effect considered | Heart changes confounding effect considered | Sample size ($n > 10$ per group) | Score | Quality criteria |
|---|---|---|---|---|---|---|---|---|
| *Takeuchi et al. (2016)* | 1 | 0 | 1 | 0 | 0 | 1 | 3 | Medium quality |
| *Thumm et al. (2018)* | 1 | 0 | 1 | 0 | 0 | 1 | 3 | Medium quality |
| *Verghese et al. (2017)* | 1 | 1 | 1 | 0 | 1 | 1 | 5 | High quality |

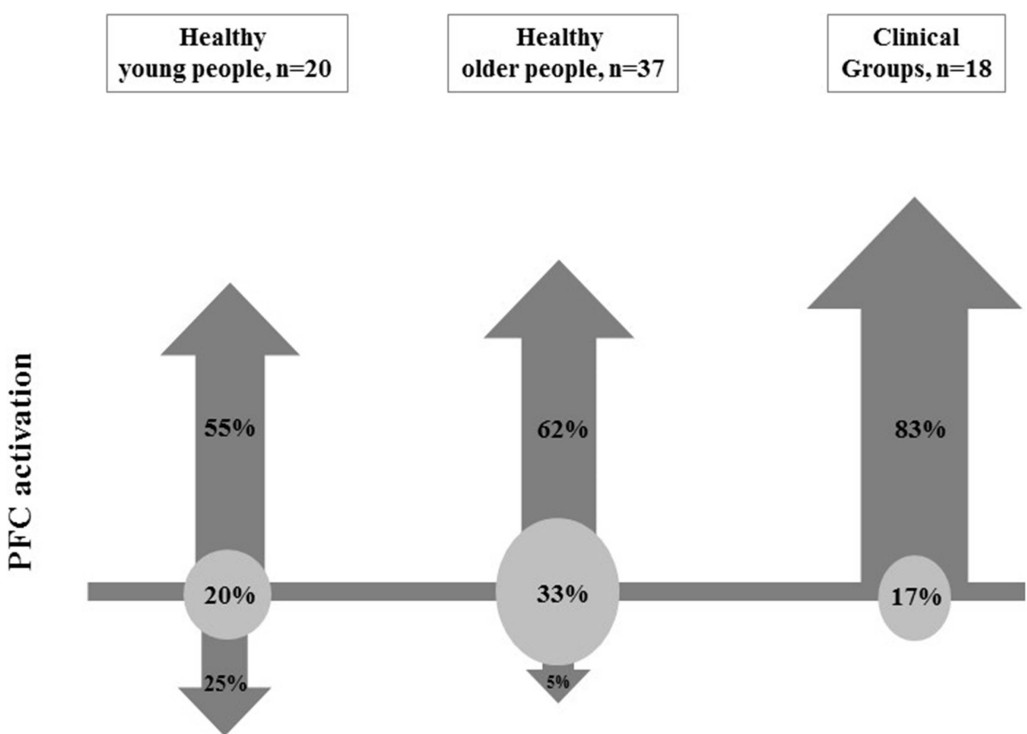

**Figure 2** Proportion of comparisons within each group showing increase (up arrows), decrease (down arrows) or no change (circle) in Prefrontal Cortex (PFC) activation when comparing walking with secondary task versus baseline.

increased cortical activation in the SMA (*Lu et al., 2015*), while one found no change in the SMA as well as no change in brain activation in the preSMA and motor and sensorimotor areas (Primary Motor Cortex (M1) and Primary Sensorimotor Cortex (S1)) (*Koenraadt et al., 2014*). One comparison also reported increased cortical activation in the PMC (*Lu et al., 2015*) while another found no change in activation in the PMC and the SMC (*Suzuki et al., 2004*). Of the 37 comparisons from 22 studies conducted in older adults, 23 comparisons reported significant increases in PFC activation with increasing locomotor task complexity (*Lucas et al., 2018*; *Mirelman et al., 2017*; *Hawkins et al., 2018*; *Chaparro et al., 2017*; *Chen et al., 2017*; *Harada et al., 2009*; *Holtzer et al., 2011*; *Holtzer et al., 2015*; *Holtzer et al., 2016*; *Holtzer et al., 2017a*; *Holtzer et al., 2017b*; *Maidan et al., 2016*; *Verghese et al., 2017*; *Clark et al., 2014a*; *Clark et al., 2014b*; *Osofundiya et al., 2016*; *Hernandez et al., 2016*); two comparisons reported a reduction in PFC activation (*Beurskens et al., 2014*; *Clark et al., 2014a*) and twelve comparisons reported no change (*Hawkins et al., 2018*; *Eggenberger et al., 2016*; *Mori, Takeuchi & Izumi, 2018*; *Harada et al., 2009*; *Takeuchi et al., 2016*; *Clark et al., 2014b*; *Maidan et al., 2016*; *Caliandro et al., 2015*; *Al-Yahya et al., 2016*). Only one study investigated other cortical areas, reporting no change in pre-SMA or Medial Sensorimotor Cortex (mSMC) activation and an increase in SMA activation (*Harada et al., 2009*). Finally, of the 18 comparisons from twelve studies conducted in clinical groups, fifteen comparisons reported increased PFC activation (*Thumm et al., 2018*; *Hawkins et al.,*

*2018*; *Chaparro et al., 2017*; *Takeuchi et al., 2016*; *Osofundiya et al., 2016*; *Hernandez et al., 2016*; *Holtzer et al., 2016*; *Maidan et al., 2016*; *Caliandro et al., 2015*; *Al-Yahya et al., 2016*; *Doi et al., 2013*) and three comparisons found no change (*Mori, Takeuchi & Izumi, 2018*; *Maidan et al., 2016*).

There was no indication that the PFC activation was associated with methodological reporting scores; i.e., increased activation was reported in 23/36 (64%) comparisons in high quality studies, 14/18 (78%) comparisons in medium quality studies and 10/19 (53%) comparisons in low quality studies; ($\chi^2 = 2.56$, $df = 2$, $p = 0.279$) (Table 4). Regarding the effects of a secondary task during walking on PFC activation, 9/14 (64%) comparisons that used counting backwards reported increases, 20/24 (83%) that used verbal fluency reported increases, 11/19 (58%) that used complex motor tasks reported increases and 0/4 (0%) that used visual tasks reported increases (Table 5).

Table 6 shows the effect of an additional motor or cognitive task relative to simple walking on gait outcomes in healthy young adults, healthy older adults and clinical groups with balance disorders. Reduced gait speed was reported in all studies investigating overground walking with the exception of one study that observed no changes in gait speed when older people walked while counting backwards by 3 or negotiated an obstacle course (*Mirelman et al., 2017*). No changes in gait speed were also observed in the two studies of treadmill walking where walking speed was controlled by the examiner (*Clark et al., 2014a*; *Clark et al., 2014b*). Shorter step/stride length was observed in all studies conducted on level surfaces except for one study that investigated counting backwards from a 3-digit number while walking on a treadmill (*Al-Yahya et al., 2016*) where both older people and stroke survivors exhibited increased stride length compared with simple walking. As expected, higher spatiotemporal variability was observed when people performed a secondary motor task that manipulated spatiotemporal characteristics, such as obstacle crossing and precision stepping (*Koenraadt et al., 2014*; *Clark et al., 2014b*; *Mirelman et al., 2017*), but also in one study where older people performed a verbal fluency task (*Clark et al., 2014a*). No changes in gait variability were observed when young people walked while performing arithmetic tasks (*Lu et al., 2015*; *Meester et al., 2014*), when young people walked carrying a tray (*Lu et al., 2015*), when somatosensory information was manipulated in older people (*Clark et al., 2014a*; *Clark et al., 2014b*) and when young and older people walked while negotiating obstacles or while counting backwards by 3 (*Mirelman et al., 2017*). One study, by *Nieuwhof et al. (2016)*, showed decreased stride length variability in people with PD when performing the digit span task while walking.

Table 7 presents group comparisons with respect to brain activation changes resulting from undertaking a complex walk between (i) healthy older and young adults and (ii) clinical groups with balance disorders and healthy peers. Of the five studies that contrasted PFC activation changes when conducting a dual task walk between young and older adults, only one study showed greater increases in PFC activation in older adults in all tasks performed (*Mirelman et al., 2017*). Another study (*Hawkins et al., 2018*), reported greater PFC activation in older people in simple walking in only the first half of an obstacle negotiation task and not when walking and performing a verbal fluency task. Two other studies reported no group differences (*Beurskens et al., 2014*; *Takeuchi et al., 2016*) and

Pelicioni et al. (2019), *PeerJ*, DOI 10.7717/peerj.6833

**Table 4  Prefrontal cortical activation in relation to methodological reporting scale.**

| | Increase | | | No change | | | Decrease | | |
|---|---|---|---|---|---|---|---|---|---|
| | low quality | medium quality | high quality | low quality | medium quality | high quality | low quality | medium quality | high quality |
| **Healthy young adults** | Suzuki et al. (2008) | Hill et al. (2013) | Lu et al. (2015) | Hawkins et al. (2018) | Takeuchi et al. (2016) | Koenraadt et al. (2014) | Hawkins et al. (2018) | Lin & Lin (2016) | Koenraadt et al. (2014) |
| | Suzuki et al. (2004) | Mirelman et al. (2014) | Lu et al. (2015) | | | Beurskens et al. (2014) | Hawkins et al. (2018) | Lin & Lin (2016) | |
| | | Lin & Lin (2016) | Maidan et al. (2018) | | | | | | |
| | | | Mirelman et al. (2017) | | | | | | |
| | | | Mirelman et al. (2017) | | | | | | |
| | | | Holtzer et al. (2011) | | | | | | |
| **Healthy older adults** | Clark et al. (2014a) | Holtzer et al. (2017a) | Holtzer et al. (2015) | Clark et al. (2014b) | Takeuchi et al. (2016) | Caliandro et al. (2015) | Clark et al. (2014a) | | Beurskens et al. (2014) |
| | Clark et al. (2014b) | Holtzer et al. (2017b) | Mirelman et al. (2017) | Hawkins et al. (2018) | | Beurskens et al. (2014) | | | |
| | Clark et al. (2014b) | Osofundiya et al. (2016) | Mirelman et al. (2017) | Hawkins et al. (2018) | | Al-Yahya et al. (2016) | | | |
| | Hawkins et al. (2018) | Osofundiya et al. (2016) | Lucas et al. (2018) | Hawkins et al. (2018) | | Harada et al. (2009) | | | |
| | Hawkins et al. (2018) | Chaparro et al. (2017) | Harada et al. (2009) | | | Maidan et al. (2016) | | | |
| | | | Chen et al. (2017) | | | Eggenberger et al. (2016) | | | |
| | | | Hernandez et al. (2016) | | | Mori, Takeuchi & Izumi (2018) | | | |
| | | | Holtzer et al. (2011) | | | | | | |
| | | | Holtzer et al. (2016) | | | | | | |
| | | | Maidan et al. (2016) | | | | | | |
| | | | Verghese et al. (2017) | | | | | | |

**Table 4** (*continued*)

| | Increase | | | No change | | | Decrease | | |
|---|---|---|---|---|---|---|---|---|---|
| | low quality | medium quality | high quality | low quality | medium quality | high quality | low quality | medium quality | high quality |
| | *Hawkins et al. (2018)* | *Osofundiya et al. (2016)* | *Caliandro et al. (2015)* | *Hawkins et al. (2018)* | | *Maidan et al. (2016)* | | | |
| | *Hawkins et al. (2018)* | *Chaparro et al. (2017)* | *Al-Yahya et al. (2016)* | | | *Mori, Takeuchi & Izumi (2018)* | | | |
| | *Hawkins et al. (2018)* | *Thumm et al. (2018)* | *Maidan et al. (2016)* | | | | | | |
| Clinical groups with balance disorders | | *Thumm et al. (2018)* | *Hernandez et al. (2016)* | | | | | | |
| | | *Nieuwhof et al. (2016)* | *Holtzer et al. (2016)* | | | | | | |
| | | *Osofundiya et al. (2016)* | *Nieuwhof et al. (2016)* | | | | | | |

**Table 5  Prefrontal cortical activation in relation to complex walking tasks.**

| | Increase | | | | No change | | | | Decrease | | | |
|---|---|---|---|---|---|---|---|---|---|---|---|---|
| | Counting backwards | Verbal fluency | Motor task | Visual task | Counting backwards | Verbal fluency | Motor task | Visual task | Counting backwards | Verbal fluency | Motor task | Visual task |
| Healthy young adults | Hill et al. (2013)<br>Lu et al. (2015)<br>Meester et al. (2014)<br>Mirelman et al. (2014)<br>Mirelman et al. (2017) | Holtzer et al. (2011) | Lu et al. (2015)<br>Suzuki et al. (2004)<br>Maidan et al. (2018)<br>Mirelman et al. (2017) | | | Beurskens et al. (2014)<br>Hawkins et al. (2018) | Koenraadt et al. (2014) | Beurskens et al. (2014)<br>Takeuchi et al. (2016) | Lin & Lin (2016) | | Koenraadt et al. (2014)<br>Lin & Lin (2016)<br>Hawkins et al. (2018) | |
| Healthy older adults | Maidan et al. (2016)<br>Mirelman et al. (2017) | Chaparro et al. (2017)<br>Chen et al. (2017)<br>Hernandez et al. (2016)<br>Holtzer et al. (2011)<br>Holtzer et al. (2015)<br>Holtzer et al. (2016)<br>Holtzer et al. (2017a)<br>Holtzer et al. (2017b)<br>Osofundiya et al. (2016)<br>Verghese et al. (2017)<br>Clark et al. (2014a)<br>Clark et al. (2014b)<br>Lucas et al. (2018) | Osofundiya et al. (2016)<br>Harada et al. (2009)<br>Clark et al. (2014b)<br>Mirelman et al. (2017) | | Al-Yahya et al. (2016)<br>Mori, Takeuchi & Izumi (2018) | Hawkins et al. (2018)<br>Beurskens et al. (2014) | Eggenberger et al. (2016)<br>Hawkins et al. (2018)<br>Maidan et al. (2016)<br>Harada et al. (2009) | Takeuchi et al. (2016) | | | | Beurskens et al. (2014) |
| | Al-Yahya et al. (2016) | Hawkins et al. (2018) | Osofundiya et al. (2016) | | Maidan et al. (2016) | | | | | | | |

Pelicioni et al. (2019), *PeerJ*, DOI 10.7717/peerj.6833

Pelicioni et al. (2019), *PeerJ*, DOI 10.7717/peerj.6833

**Table 5** (*continued*)

| | Increase | | | | No change | | | | Decrease | | | |
|---|---|---|---|---|---|---|---|---|---|---|---|---|
| | Counting backwards | Verbal fluency | Motor task | Visual task | Counting backwards | Verbal fluency | Motor task | Visual task | Counting backwards | Verbal fluency | Motor task | Visual task |
| Clinical groups with balance disorders | *Nieuwhof et al. (2016)* | *Chaparro et al. (2017)* | *Hawkins et al. (2018)* | | *Mori, Takeuchi & Izumi (2018)* | | | | | | | |
| | | *Hernandez et al. (2016)* | *Maidan et al. (2016)* | | | | | | | | | |
| | | *Doi et al. (2013)* | | | | | | | | | | |
| | | *Holtzer et al. (2016)* | | | | | | | | | | |
| | | *Osofundiya et al. (2016)* | | | | | | | | | | |

**Table 6  Effect of the additional tasks on gait outcomes compared to simple walking in healthy young adults, healthy older adults and clinical groups with balance disorders.**

| | Healthy young adults | Healthy older adults | Clinical groups with balance disorders |
|---|---|---|---|
| *Motor secondary task* | | | |
| *Chen et al. (2017)* | | Gait speed (−) | |
| *Clark et al. (2014b)* | | Gait speed (−), step length variability (+)[a] | |
| *Hawkins et al. (2018)* | Gait speed (−) | Gait speed (−) | Gait speed (−) |
| *Koenraadt et al. (2014)* | Step time variability (+) | | |
| *Lin & Lin (2016)* | Gait speed (−), step length (−) | | |
| *Lu et al. (2015)* | Gait speed (−), cadence (=), stride length (−), gait variability (=) | | |
| *Mirelman et al. (2017)* | Gait speed (−), stride length (+), gait variability (+) | Gait speed (=), stride length (+), gait variability (+) | |
| *Osofundiya et al. (2016)* | | Gait speed (−) | Gait speed (−) |
| *Somatossensory task* | | | |
| *Clark et al. (2014a)* | | step length (=)[b], gait speed (=)[b], step length (−)[c], gait speed (−)[c], step length variability (=)[b,c] | |
| *Clark et al. (2014b)* | | gait speed (=), step length variability (=) | |
| *Chaparro et al. (2017)* | | Stride length (−) | Stride length (−) |
| *Cognitive secondary task* | | | |
| *Al-Yahya et al. (2016)* | | Stride length (+), cadence (−) | Stride length (+), cadence (−) |
| *Beurskens et al. (2014)* | Step length (−) | Step length (−) | |
| *Chaparro et al. (2017)* | | Stride length (=) | Stride length (=) |
| *Clark et al. (2014a)* | | step length (−)[b,c], step length variability (+)[b,c], gait speed (=)[b], gait speed (−)[c] | |
| *Clark et al. (2014b)* | | Gait speed (−), step length variability (=) | |
| *Doi et al. (2013)* | | | Gait speed (−) |
| *Hernandez et al. (2016)* | | Gait speed (−) | Gait speed (−) |
| *Hill et al. (2013)* | Gait speed (−) | | |
| *Holtzer et al. (2011)* | Gait speed (−) | Gait speed (−) | |
| *Holtzer et al. (2015)* | | Stride length (−), gait speed (−) | |
| *Holtzer et al. (2016)* | | Gait speed (−) | Gait speed (−) |
| *Holtzer et al. (2017a)* | | Gait speed (−) | |
| *Holtzer et al. (2017b)* | | Gait speed (−) | |

**Table 6 (*continued*)**

| | Healthy young adults | Healthy older adults | Clinical groups with balance disorders |
|---|---|---|---|
| *Lin & Lin (2016)* | Gait speed (−), step length (−) | | |
| *Lu et al. (2015)* | Gait speed (−), cadence (−), stride length (−), gait variability (=) | | |
| *Lucas et al. (2018)* | | Gait speed (−) | |
| *Meester et al. (2014)* | Step time variability (=) | | |
| *Mirelman et al. (2014)* | Stride length (−)[d], gait speed (−) | | |
| *Mirelman et al. (2017)* | Gait speed (−),stride length (=), gait variability (=) | Gait speed (=),stride length (=), gait variability (=) | |
| *Nieuwhof et al. (2016)* | | | Stride length (−) (e ), Stride length variability (−)[e] |
| *Osofundiya et al. (2016)* | | Gait speed (−) | Gait speed (−) |
| *Verghese et al. (2017)* | | Gait speed (−) | |

**Notes.**
(+), increase of spatiotemporal parameter when performing the additional task during walking; (−), decrease of spatiotemporal parameter when performing the additional task; (=), no changes in spatiotemporal parameter.
Significant differences only for:
[a]Obstacle negotiation.
[b]Treadmill walking.
[c]Overground walking.
[d]Walking + subtracting by 7s.
[e]Walking while reciting digit spans.

**Table 7  Prefrontal cortical activation pattern differences between healthy young and healthy older adults and between clinical groups with balance disorders and healthy peers.**

| | PFC activity |
|---|---|
| **Healthy older vs. healthy young adults** | |
| *Beurskens et al. (2014)* | **DT (=)** |
| *Holtzer et al. (2011)* | **DT (−)** |
| *Takeuchi et al. (2016)* | **DT (=)** |
| *Hawkins et al. (2018)* | **SW (+)/OBS a (+)/OBS b (=)/DT (=)** |
| *Mirelman et al. (2017)* | **SW (+)/OBS (+)/DT (+)** |
| **Clinical group with balance disorders vs. healthy peers** | |
| *Al-Yahya et al. (2016)* (Stroke) | **DT (=)** |
| *Chaparro et al. (2017)* (Multiple Sclerosis) | **DT (+)/PB (−)** |
| *Hawkins et al. (2018)* (Stroke) | **SW (=)/OBS a (=)/OBS b (+)/DT (=)** |
| *Hernandez et al. (2016)* (Multiple Sclerosis) | **DT (+)** |
| *Holtzer et al. (2016)* (Neurological gait) | **DT (=)** |
| *Maidan et al. (2016)* (Parkinson's disease) | **DT (−)/OBS (=)** |
| *Mori, Takeuchi & Izumi (2018)* (Stroke) | **DT (−)** |
| *Osofundiya et al. (2016)* (Obesity) | **DT (=)/PS (=)** |

**Notes.**
DT, Dual-task; SW, Simple walking; OBS, Obstacle negotiation; a, first half of the task; b, second half of the task; PB, Partial body support; PS, Precision stepping; (+), higher activation when performing the additional task; (−), lower activation when performing the additional task; (=), no change in activation.

one reported a relatively smaller increase in PFC activation in older people (*Holtzer et al., 2011*).

Thirteen comparisons from eight studies have contrasted PFC activation changes when conducting a dual task walk between clinical groups with balance disorders and healthy peers. Four of these reported a relatively larger increase in PFC activation in clinical groups with balance disorders. Two comparisons showed increased PFC activation when people with multiple sclerosis performed cognitive dual tasks (*Chaparro et al., 2017*; *Hernandez et al., 2016*), while the other comparisons showed increased PFC activation when stroke survivors walked while performing a cognitive dual task (*Al-Yahya et al., 2016*) or in the first half of an obstacle negotiation task (*Hawkins et al., 2018*). Six comparisons reported no between-group PFC activation differences. In three of these comparisons, stroke survivors (*Hawkins et al., 2018*), PD (*Maidan et al., 2016*) and obesity (*Osofundiya et al., 2016*) performed motor tasks (people with obstacle negotiation and precision stepping group), while in three other comparisons, stroke survivors (*Hawkins et al., 2018*), people with neurological gait (*Holtzer et al., 2016*) and obesity group (*Osofundiya et al., 2016*) performed cognitive tasks. Finally, the three reports of PFC decreases in clinical groups were observed when people with multiple sclerosis walked with partial body support (*Chaparro et al., 2017*) and when people with PD and stroke survivors performed a cognitive dual task (*Maidan et al., 2016*; *Mori, Takeuchi & Izumi, 2018*).

## DISCUSSION

This systematic review summarizes the published findings regarding PFC cortical patterns of activation in healthy young adults, healthy older adults and clinical groups with balance disorders, to gain an insight into neural processes during simple and complex walking tasks. Approximately 60% of the study comparisons reported that healthy young and older adults exhibited higher PFC activation when performing a complex task while walking compared with a baseline simple walking task; this was also the case for more than 80% of study comparisons of clinical groups with balance disorders. Moreover, PFC activation appears to be related to the type of complex walk undertaken.

### Brain activation in healthy young adults

Compared with simple walking, PFC activation increased when individuals performed: (i) fast walks (*Suzuki et al., 2004*); (ii) negotiated expected obstacles (*Mirelman et al., 2017*; *Maidan et al., 2018*) and unexpected obstacles of different heights (*Maidan et al., 2018*); (iii) performed a secondary task while walking (*Holtzer et al., 2011*; *Mirelman et al., 2014*); the last being the paradigm used in most studies that have assessed brain activation patterns in healthy young adults. In these studies, participants performed the following secondary tasks: subtracting numbers by 1, 3 or 7 (*Hill et al., 2013*; *Meester et al., 2014*; *Mirelman et al., 2014*; *Mirelman et al., 2017*), walking while talking (*Holtzer et al., 2011*) and talking and carrying a bottle of water on a tray (*Lu et al., 2015*). However, increased hemodynamic responses in the PFC appear to be task-specific. Indeed, several studies of healthy young adults (*Hawkins et al., 2018*; *Koenraadt et al., 2014*; *Lin & Lin, 2016*; *Beurskens et al., 2014*; *Takeuchi et al., 2016*) reported no change or even a decrement in PFC activation during

either secondary motor task performance (e.g., precision stepping, crossing obstacles or walking on a narrow pathway) or cognitive task performance (e.g., visual checking, alphabet recall, memory task and manipulating a smartphone). Such findings suggest that increasing balance/ locomotor changes does not require additional PFC activation in healthy young adults, and might involve other cortical and subcortical areas involved in the control of locomotion. In addition, secondary tasks involving working memory (*Lin & Lin, 2016*; *Beurskens et al., 2014*) might also involve cortical areas in addition to the PFC that were not investigated in the published studies.

## Brain activation in healthy older adults

Most studies (*Lucas et al., 2018*; *Mirelman et al., 2017*; *Hawkins et al., 2018*; *Chaparro et al., 2017*; *Chen et al., 2017*; *Holtzer et al., 2011*; *Holtzer et al., 2015*; *Holtzer et al., 2016*; *Holtzer et al., 2017a*; *Holtzer et al., 2017b*; *Clark et al., 2014a*; *Clark et al., 2014b*; *Osofundiya et al., 2016*; *Hernandez et al., 2016*; *Maidan et al., 2016*; *Verghese et al., 2017*) but not all (*Al-Yahya et al., 2016*; *Beurskens et al., 2014*; *Takeuchi et al., 2016*; *Hawkins et al., 2018*; *Eggenberger et al., 2016*; *Harada et al., 2009*; *Mori, Takeuchi & Izumi, 2018*); reported that when healthy older adults performed cognitive or motor tasks while they walked, the PFC was more activated in comparison to baseline conditions. It seems that in healthy older adults, PFC activation increases with secondary cognitive tasks that involve attention and executive functioning (e.g., walking while subtracting). In contrast, and similar to what is noted in healthy young adults, the conduct of tasks that require speed manipulation (*Harada et al., 2009*; *Eggenberger et al., 2016*), visual checking (*Beurskens et al., 2014*), unpractised tasks (manipulating a smartphone) (*Takeuchi et al., 2016*), or obstacle negotiation (*Hawkins et al., 2018*; *Maidan et al., 2016*) do not appear to increase PFC activation. As with young adults, increased activation may occur in other cortical areas that process visual-spatial stimuli (*Wu et al., 2018*).

## Brain activation in healthy individuals: age comparisons

Healthy older adults usually walk slower and have more difficulty performing dual tasks than healthy young adults, (*Al-Yahya et al., 2011*). However, of the five studies that investigated between age-group effects when performing walks with secondary tasks, only one study observed greater PFC activation in older people performing different secondary tasks (obstacle negotiation and counting backwards) (*Mirelman et al., 2017*). PFC activation patterns were not different between healthy young and older adults in three studies (*Beurskens et al., 2014*; *Takeuchi et al., 2016*; *Hawkins et al., 2018*) and lower in older adults in one study (*Holtzer et al., 2011*). The limited number of studies which have explored the effects of aging as well as the nature of the secondary task used in these studies might account for the lack of an age effect.

## Brain activation in clinical groups with balance disorders

In most clinical groups with balance disorders, including stroke survivors (*Al-Yahya et al., 2016*; *Hawkins et al., 2018*), obese individuals (*Osofundiya et al., 2016*), individuals with ataxia (*Caliandro et al., 2015*), multiple sclerosis (*Chaparro et al., 2017*; *Hernandez et al., 2016*), peripheral neuropathy (*Holtzer et al., 2016*), and mild cognitive impairment (*Doi et*

*al., 2013*), higher PFC activation has been reported regardless of the type of concomitant task performed during ambulation. This is also the case for comparisons made in studies in people with PD (*Maidan et al., 2016*; *Thumm et al., 2018*; *Nieuwhof et al., 2016*) with one exception; *Maidan et al. (2016)* found no change in PFC activation when comparing walking whilst performing a concomitant subtracting task with simple walking.

## Theoretical considerations

Our findings of enhanced hemodynamic responses in the PFC apparent when older adults and individuals with balance disorders perform complex walking tasks align particularly with the notion that increased cortical activity reflects a compensatory mechanism (*Cabeza et al., 2002*; *Reuter-Lorenz & Cappell, 2008*; *Grady, 2012*). This might reflect the need to allocate more attentional resources to walking while performing secondary tasks, or the need to use a more direct locomotor pathway due to deficits in automaticity (e.g., as generally observed in individuals with PD) (*Herold et al., 2017*). The age-related differences are also consistent with the frontal lobe hypothesis of aging (*West, 1996*) and the cognitive reserve theory which supports that older adults increase brain activity by a larger degree to cope with elevated cognitive task difficulty (*Stern, 2009*). Moreover, these functional effects of aging mirror age-related structural changes proposed by the "last-in-first-out" hypothesis where late maturing brain regions decline first in later life (*Raz & Kennedy, 2009*; *Tamnes et al., 2013*; *Bender, Volkle & Raz, 2015*) and explain gait disturbances.

## Differential effects of secondary task type on PFC activation

Four secondary task types were commonly used in the included studies: counting backwards, verbal fluency, motor tasks and visual tasks. Of these, verbal fluency was the most consistent in increasing PFC activation (*Lucas et al., 2018*; *Hawkins et al., 2018*; *Chaparro et al., 2017*; *Chen et al., 2017*; *Holtzer et al., 2011*; *Holtzer et al., 2015*; *Holtzer et al., 2016*; *Holtzer et al., 2017a*; *Holtzer et al., 2017b*; *Clark et al., 2014a*; *Clark et al., 2014b*; *Osofundiya et al., 2016*; *Hernandez et al., 2016*; *Verghese et al., 2017*; *Doi et al., 2013*). Counting backwards also increased PFC in most studies (*Lu et al., 2015*; *Hill et al., 2013*; *Meester et al., 2014*; *Mirelman et al., 2014*; *Mirelman et al., 2017*; *Maidan et al., 2016*; *Al-Yahya et al., 2016*; *Nieuwhof et al., 2016*). Of the 9 comparisons that did not show increased PFC activation with verbal fluency or backward counting tasks, some were from studies of low (*Hawkins et al., 2018*) or medium methodological reporting quality (*Lin & Lin, 2016*) that did not report control for motion artefacts, external lighting and physiological noise, and in two studies (*Beurskens et al., 2014*; *Al-Yahya et al., 2016*) the secondary tasks were performed on a treadmill. This may be important as *Clark et al. (2014a)* observed that when participants walked on treadmill no change in PFC activation was observed between dual-task (verbal task) and baseline condition, while changes were observed on overground walking.

Half of the studies that examined the effects of motor tasks during walking in healthy individuals found increases in PFC. In the studies where PFC was not increased, three were performed on a treadmill (*Eggenberger et al., 2016*; *Harada et al., 2009*; *Koenraadt et al., 2014*), one was of low methodological reporting quality (*Hawkins et al., 2018*), one did not

control for important aspects that could have affected the interpretation of the data, such as motion artefacts, external lighting and physiological noise (*Lin & Lin, 2016*), and one was conducted in people with PD (*Maidan et al., 2016*). Obstacle negotiation and precision stepping tasks increased PFC in the three studies performed in clinical groups with balance disorders (*Osofundiya et al., 2016*; *Maidan et al., 2016*; *Hawkins et al., 2018*), but this may simply indicate any additional load may elicit such changes in such populations.

Finally, visual tasks such as visual checking and manipulating a smartphone did not increase PFC activation (*Beurskens et al., 2014*; *Takeuchi et al., 2016*). For these task types, other brain areas such as the visual cortex might be more involved. However, further studies directly assessing visual cortex activation as well as other cortical regions are required to confirm this hypothesis.

## Study limitations

Studies addressing gait with fNIRS are still at a relatively early stage, with best practice methodology evolving as experience with this technique is garnered. Studies may have not met particular quality criteria due to the pioneering nature of the studies using this new technology and/or omission of reporting of all methodological factors and most of the papers (89%) had one or more of the following methodological limitations: small sample sizes, no indication of removal or control of motion artefacts or physiological noise in data processing and sub-optimal number and positioning of optodes. Further, the secondary tasks used in many studies involved speaking (counting backwards and verbal fluency) requiring muscles adjacent to the PFC (*Zimeo-Morais et al., 2018*). Such muscle activity as well as different facial expressions (*Balardin et al., 2017*) may affect fNIRS signal quality.

To address the above, we recommend that in future studies, sample sizes be based on power analyses of expected effect sizes for spatiotemporal gait and hemodynamic measures to provide confidence in the study findings. Second, motion artefacts should be removed during the data processing or be controlled for; confounding physiological noise such as fluctuations in heart rate should either be monitored and reported, or controlled for using appropriate filtering. Third, baseline and test trials should be of sufficient duration to detect the slow changing hemodynamic signals as oxygenated blood starts flowing between 1 and 2s after stimuli onset and achieves its peak approximately 6s after stimulus onset (*Holtzer et al., 2011*). Finally, EEG standards (i.e., 10-5) and/or anatomical maps to define optode positions (i.e., Brodmann areas) for the brain regions of interest should be used. However, we acknowledge that given the limited number of fNIRS channels in current devices, this may prove to be an unavoidable limitation for some investigations (*Koenraadt et al., 2014*; *Maidan et al., 2016*).

We also acknowledge some limitations. First, there was considerable heterogeneity of study protocols in the present review. As such, variations in baseline conditions (e.g., sitting/standing/unspecified), walking speed (e.g., self-selected/controlled), duration and amount of trials, treadmill vs. overground walking, montage, inter-optode distance, etc. limited the clustering of studies and hindered the overall interpretation of the data. Finally, factors such as motor repertoire, physical activity, practice and skill levels, risk of falling and hemispheric asymmetry (*Ekkekakis, 2009*; *Erickson et al., 2007*; *Jancke, Shah & Peters,*

*2000*; *Naito & Hirose, 2014*) can affect cortical activity but were beyond the scope of this review. Complementary studies and reviews are required to elucidate the influence these factors have on cortical activity and associated balance control.

## CONCLUSION

This systematic review revealed that the majority of studies found increased PFC activation with increased walking task complexity in young and older people and clinical groups with balance disorders. However, increased PFC activation was most common in studies that contained walks comprising secondary tasks of verbal fluency, arithmetic and alphabet reciting. The finding that clinical groups with balance disorders generally showed increased PFC activation irrespective of type of secondary task during walking suggests these groups require more attentional resources for safe walking.

### Funding

Paulo Pelicioni is a recipient of a Coordenação de Aperfeiçoamento de Pessoal de Nível Superior (CAPES) PhD scholarship [Grant number: BEX 2194/15-5]. Stephen Lord is supported by an NHMRC Research Fellowship. The funders had no role in study design, data collection and analysis, decision to publish, or preparation of the manuscript.

### Grant Disclosures

The following grant information was disclosed by the authors:
Coordenação de Aperfeiçoamento de Pessoal de Nível Superior (CAPES) PhD scholarship: BEX 2194/15-5.
NHMRC Research.

### Competing Interests

The authors declare there are no competing interests.

### Author Contributions

- Paulo H.S. Pelicioni conceived and designed the experiments, performed the experiments, analyzed the data, contributed reagents/materials/analysis tools, prepared figures and tables, authored or reviewed drafts of the paper, approved the final draft.
- Mylou Tijsma performed the experiments, analyzed the data, approved the final draft.
- Stephen R. Lord conceived and designed the experiments, contributed reagents/materials/analysis tools, authored or reviewed drafts of the paper, approved the final draft.
- Jasmine Menant conceived and designed the experiments, performed the experiments, analyzed the data, contributed reagents/materials/analysis tools, authored or reviewed drafts of the paper, approved the final draft.

### Data Availability

The summary data is in Table 1.

## Supplemental Information

Supplemental information for this article can be found online at http://dx.doi.org/10.7717/peerj.6833#supplemental-information.

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
