# Peer review of "Prefrontal cortical activation measured by fNIRS during walking: effects of age, disease and secondary task"

_PeerJ, doi:10.7717/peerj.6833_

## Round 0.1 · original submission · Major Revisions

As you will see, the reviewers provided thoughtful and detailed comments and suggestions for changes that they felt would substantially improve the value of this work for readers. I therefore encourage you to give careful consideration to each of the points raised, and to revise the text and tables as appropriate. Thank you again for submitting your work to PeerJ, and I hope that this review process proves helpful to you.

·

Basic reporting

no comment

Experimental design

The authors discuss the studies based on several comparisons mentioned in the Results section. However, in the Methodology I could not find any hints regarding an exact definition of comparisons. Thus, it is not completely clear what the authors mean when they write for example “11 of the 20 comparisons from 13 studies” (l. 38, l. 189). Two comparisons relate to the comparison of the complex task to the baseline and the simple task for example? Now, it is still ambiguous for me how the authors did the comparisons. I think that the understanding of the Results and the whole manuscript would improve if the authors insert additional information on comparisons in the Methodology section.

Regarding Figure 1: Four stages were presented in the figure, but stage 1 and stage 2 were only described in the manuscript. Please provide short information to stage 3 with the final inclusion of 35 papers.

Validity of the findings

Regarding Table 1: It shows that the studies are really heterogenous regarding baseline (e.g. sitting/standing/unspecified), speed (e.g. self-selected/controlled), duration and amount of trials, treadmill vs. overground walking, montage, interoptode distance etc. Of course, it is not possible to discuss every difference in detail. Nevertheless, it is important and essential to keep in mind while reading the review – for interpreting and conclude the findings of the review and for planning further fNIRS studies. Therefore, I expect a few more sentences in the discussion and possibly in the conclusion where the heterogeneity of the state of research is mentioned.

Additional comments

General comments:
My overall impression of this systematic review is that it is well-written, good structured, and comprehensible in all parts. The research question and findings are of high relevance as it summarizes a rising number of studies about cortical activation patterns during walking. The methodological procedure is detailed and well-presented. In the discussion section, some comments should be added which are important and essential for interpreting this summary and planning further fNIRS studies. As I have no fundamental concerns regarding methodology procedure, result presentation and interpretation, I have only some additional questions and comments that might help to improve the clarity of the manuscript to get published here or elsewhere. Please see my detailed and specific comments regarding these factors below.

Specific comments:
Regarding Table 1: Be consequent in using terms like oxyHb or oxyHB (e.g. line 1, Results section).

Regarding Table 1: Please add the abbreviations in the subheading.

Regarding Table 2: Please check the information for incorrectness (e.g. Vorghese et al., 2017 used 10s standing as baseline not simple walking). The same comment applies to Table 1.

Regarding Table 3: Is there any reference for the methodological criteria? I wonder at quality criteria such as sample size. I am not really convinced because the sample size and power depend on the experimental design. Power analysis a priori with effect sizes would have been a good approach you have discussed in the study limitations. Please add that your evaluation refers only to the availability of the information, not if the studies have highly qualified experimental designs.

Regarding Table 6: Be consistent with the headings (e.g. young people in Table 2, Healthy young people in Table 6, healthy peers in Table 7).

Regarding Figure 2: The authors present the scores of the methodological quality criteria but what is the message of these data? In case you want to include this figure, I would like to know what the information report. For example, why do only 20% of the papers cover interferences with external light? Additionally, I would prefer to exclude the pie chart. On one side, it does not contain further information or clarify any results. One the other side, there are many figures and tables in the manuscript already.

Regarding Figure 3: Percentages are presented in the figure whereas numbers are presented in the manuscript. Please be consistent in the figures, tables and the manuscript.

·

Basic reporting

There are some reviews about fNIRS on brain activity during dual-tasks; but the mains idea about this review is original. However, there are many problems that we need to point out and I’m summarizing below the main problems.
1) The main idea to perform a review is summarize the data we have on the literature. I believe that the authors did not summarize in a good way. The tables and figures are huge, with much unnecessary information, information that the authors did not discuss, etc.
2) Another problem is the way you talk about prefrontal activation. To identify an increase on activation, I think that you are talking about the increase in oxy-hemoglobin. You need to clarify it, because the fNIRS can give oxy, deoxy, total hemoglobin, TOI, etc. Nevertheless, I suggest you focus on oxy-hemoglobin changes, focus the paper on this variable.

Experimental design

Abstract
I think this section could be better. The main problem is that results and conclusion do not match. is really good. So, I have some suggestions:
- On row 24, “regarding Prefrontal cortical activation (PFC)…” I suggest “regarding Prefrontal cortical (PFC) activation”.
- Results: this abstract section is really poor. The results presented do not meet with the conclusion of the abstract section. On this results, you are just showing the quality assessment of the papers; while on the conclusions you are talking about the influence of different cognitive tasks on gait, about clinical groups, etc.
- Keywords: I believe that you need to write some keywords at the end of the abstract section.

Introduction
- I suggest a reduction in some sentences. There are many long sentences that should be divided into more sentences. An example is a sentence that starts on line 51 and finishes on line 55.
- On row 59. The fNIRS is used for investigating cortical activation just while participants are moving freely? It is confusing to me. I think that you need to redo this sentence.
- On row 66, you used PFC as Prefrontal Cortex. However, you have been used as Prefrontal cortical activation. You need to adjust it.

Methods:
- On row 99 you said that you used 4 databases. However, on figure 1 you stated 12 were identified through manual search. You wrote it at the end of the methods section. I suggest you write it at the beginning because, on figure 1, you showed that you used the manual search at the beginning of the screaming.
- On the selection criteria section, on the first stage of screaming you have 8 selections criteria. But in figure 1, the excluded motivations are completely different from what you wrote in this section. I suggest you use these 8 selections criteria in figure 1 in order to explain better the exclusion motivation. Similarly it is happening on the second stage of screaming. You need to adjust it as well.
- There are 4 different languages that you selected papers. So, at list two of each author can read Portuguese, Dutch, and French, right?
- How did you define “clinical groups with balance disorders”? I believe that at the methods section you need to give to the readers one definition about this statement.
- On row 127, you wrote: “PFC activity change was the primary outcome measured in this systematic review”. However, the fNIRS can give oxy, deoxy, total hemoglobin, TOI… so, the main outcome should be one of these. In my opinion, it can not be “PFC activity”.
- Table 1: the table is giant, lots of information and so confusing. You have much information that you did not discuss in the paper.

Validity of the findings

Results:
- This section is much longer than necessary. Many times you are writing exactly what you have on tables and figures. You need to decide if you want to show on the text or on the figures or tables. Another point is that most of the tables and figures are not clear, they are big and confusing. You need to summarize these tables and figures in a lower number of tables and figures because many information is not necessary and because you did not discuss this information.
- Table 2: this table is more clear than table 1. However, the numbers that you used to indicate the tasks performed by each study make the table not easily friend when you are looking at. And, you used “clinical group” in all studies. So, if the readers want to identify what pathological group you are talking about, the readers need to search on the paper. It is a confusing way to present the results on a table.
- Figure 2: what methodological criteria did you use to classify the papers on high, medium or low quality. You need to explain it.
- Figure 3: On row 186 you started showing “an increase, decrease or no change in PFC activation”. Are you talking about an increase, decrease or no change on oxy-hemoglobin levels? I believe that you are talking about it, but you need to make it clear. It is crucial. Another important point: These increase or decrease are statistically significant? What is the “p” value? This is another important point because if it is not, doesn’t make sense to report it. This figure has other problem that you have in figure 2: you used: “clinical group” here again.
- Table 6: again; the differences are statistically significant? What is the “p” value? And again, the numbers that you used to indicate the tasks performed by each study make the table not easily friend when you are looking at.
- Table 7: again; the differences are statistically significant? What is the “p” value?

Discussion:
- Several times on this section you just repeated what you have on results or tables or figures, sometimes on results and tables or on results and figures. You need to discuss the results training to understand why you have these results. You need to avoid using “may, might” to justify the results. You need to make your ideas more strong using more papers not included in the 35 selected papers.
- Also, you need to discuss trying to understand if the papers’ results are related to the participants’ age, sample size, etc.
- On row 325, you wrote, “However, increased hemodynamic responses in the PFC appear to be task-specific”. But, how do you classify easy and difficult tasks? For example, why did you consider “crossing obstacles” as an easy task? And, you did not differ cognitive tasks than motor tasks on this statement. To make a conclusion like this one, you need to discuss better the results you have, to show differences between easy and difficult tasks, etc.
- On row 333, you stated, “secondary tasks involving working memory might also involve cortical areas other than the PFC that were not investigated in the published studies”. But if it is involving other cortical areas doesn’t mean that PFC is not involving; and, there are many studies using working memory to analyze differences on PFC activation.
- On row 343; “the PFC was more activated…” compared to what? Single task, young people?
- On row 350, “As with young adults, increased activation may occur in other cortical areas that process visual-spatial stimuli”. You need to use more references to make this statement. Using “might or may” without references supporting the statements make your statement weak.
- On row 366, I saw that you used a study with obese people. Put together in the same “clinical group”, for example, people with stroke and obese people is not a better way to discuss the results.
- On row 373, you wrote a long sentence speculating about hemodynamic responses, using “may” again without any reference.
- On row 380. You need to define what is a difficult task and why it is.

Conclusions:
- The way you presented the results and discussed the paper did not allow you to make the conclusions you did.

Reviewer 3 ·

Basic reporting

This is an important and methodologically well performed systematic review on an important emerging research topic that will be of interest to the readers of this journal. Overall, this review of the relevant literature is thorough. It evaluates prefrontal cortical activation patterns change when people perform gait tasks of increasing complexity requiring somatosensory, motor or cognitive tasks, it compares young and old and patient groups with healthy controls, and, finally, assesses the quality of existing studies investigating prefrontal cortical activation measured by fNIRS during walking considering the effects of age, disease and secondary task(s).
The Introduction and Discussion sections are well structured, however, could be more informative. What were, for example, the expectations for the cortical activation pattern differences and/or changes and why? It is clear that such an underlying rationale cannot be derived from existing fNIRS works because of a dearth of information; however, brain cortical activation as assessed with other methodologies might be used for this purpose. This could give readers important hints leading to the underlying hypotheses driving this review. What is missing from the paper in the Discussion is a section of the paper where the findings are set in the context of existing works and where possible explanatory hypotheses are given for the observations. This information should be inserted since this will be able giving direction for future studies. A systematic review should not only provide readers a quick way to determine which papers they can trust based on for example a quality assessment, however, they should also offer readers contextual information what those papers are telling us by interpreting the findings and putting these in a context of existing works. They allow us to get a big picture of what is known in a field by doing this.
I do not see obvious issues with methodology or conclusions, but offer a few suggestions in hopes that the authors might find them helpful in revising their manuscript.

The authors should give some information what decreased/increased cortical activation related to walking is thought to represent. From other works we generally assume that decreased activity of a particular brain area may represent decreased use and, therefore, increased efficiency; see for a discussion on this the reviews by Lustig, Shah et al. (2009) and Grady (2012). Some sources use the “compensation hypothesis” as a theoretical explanatory model. In this model additional brain activity in older compared to younger adults was suggested to reflect a compensatory mechanism (compensation hypothesis) (Cabeza, Anderson et al. 2002, Reuter-Lorenz and Cappell 2008) applied to improve performance in a specific task and such over-recruitment might as well be associated with less efficient use of neural resources (Grady 2012). This relation has not only been observed between young and old adults but also between higher fit and lower fit old adults. In the current review the main focus is on the effects of age, disease and secondary task(s); however, why are other possible biasing factors such as level of fitness, physical activity, etc. currently neglected? This should be explained or discussed in the limitations of the study section.
Decreased activity of prefrontal areas are discussed as possibly resulting from a shift from controlled gait to more automatic gait or a shift from the indirect to the direct locomotor pathway, respectively (Hamacher, Herold et al. 2015). This makes it obvious to also focus on this aspect when studying fNIRS while walking and observing the seemingly contradictory findings as presented in this review. While the indirect locomotor pathway regulates gait via prefrontal cortex, premotor area, supplementary motor area, and basal ganglia, the direct pathway comprises primary motor cortex (M1), cerebellum, and spinal cord (la Fougere, Zwergal et al. 2010, Zwergal, Linn et al. 2012). Similar observations of reduced brain activity have also been reported in studies with highly skilled, professional pianists (Jancke, Shah et al. 2000) or soccer players (Naito and Hirose 2014) that both showed lower recruitment of motor areas in which they have richer sensory-motor experiences (finger or foot movements, respectively) compared to amateurs.
From the literature we can also derive behavioral differences between hemispheres regarding activation levels under different task constraints. For example in the paper of Eggenberger and colleagues, included in this review, the comparison of brain activity in the left versus the right PFC showed emerging hemispheric differences at the end of training. Such a finding needs to be discussed in the context of this review by assessing which part(s) of the PFC were actually studied in the included manuscripts (left, right, left and right) and whether the localization of the assessment together with the demographics of the investigated participants (e.g., their fitness level) may or may not account for some of the perhaps unexpected findings. According to the review by Ekkekakis (2009), prefrontal hemispheric asymmetry has not yet been investigated with fNIRS in relation to exercise. However, the observation from Eggenberger et al. seems in line with a cognitive training fMRI study that found increased prefrontal hemispheric asymmetry during dual-task conditions (Erickson, Colcombe et al. 2007). Similarly, Bergerbest, Gabrieli et al. (2009) demonstrated in their fMRI study on implicit memory (repetition priming) an initial bilateral PFC activity in older adults and left lateralized activity in younger adults. These initial activity patterns were followed by repetition related activity reductions, which were smaller for the older compared to the younger adults in the left PFC but larger in the right PFC. These latter findings would correspond with the complementary hypothesis (Colcombe, Kramer et al. 2005), proposing that in older adults bilateral brain activity or a reduction of asymmetric brain activity is not generally related to better performance in a certain task, as it is suggested by the compensation hypothesis (Cabeza, Anderson et al. 2002, Reuter-Lorenz and Cappell 2008).

Other theories that should be reflected on because some of the results of the systematic review are in line with them, are the frontal lobe hypothesis of aging (West 1996) and the cognitive reserve theory which assumes that younger adults increase brain activity by a larger degree to cope with elevated cognitive task difficulty (Stern 2009). Notably, the age-related neural modulation patterns of functional over- and under-recruitment are emerging especially within the transition from middle-aged to old adults (Kennedy, Rodrigue et al. 2015). Moreover, these functional effects of aging are mirroring age-related structural effects proposed by the “last-in-first-out” hypothesis where late maturing brain regions decline first in later life (Raz and Kennedy 2009, Tamnes, Walhovd et al. 2013, Bender, Volkle et al. 2015) and explain gait disturbances.

References used for this peer review
Bender, A. R., M. C. Volkle and N. Raz (2015). "Differential aging of cerebral white matter in middle-aged and older adults: A seven-year follow-up." Neuroimage 125: 74-83.
Bergerbest, D., J. D. Gabrieli, S. Whitfield-Gabrieli, H. Kim, G. T. Stebbins, D. A. Bennett and D. A. Fleischman (2009). "Age-associated reduction of asymmetry in prefrontal function and preservation of conceptual repetition priming." Neuroimage 45(1): 237-246.
Cabeza, R., N. D. Anderson, J. K. Locantore and A. R. McIntosh (2002). "Aging gracefully: compensatory brain activity in high-performing older adults." Neuroimage 17(3): 1394-1402.
Colcombe, S. J., A. F. Kramer, K. I. Erickson and P. Scalf (2005). "The implications of cortical recruitment and brain morphology for individual differences in inhibitory function in aging humans." Psychol Aging 20(3): 363-375.
Ekkekakis, P. (2009). "Illuminating the black box: investigating prefrontal cortical hemodynamics during exercise with near-infrared spectroscopy." J Sport Exerc Psychol 31(4): 505-553.
Erickson, K. I., S. J. Colcombe, R. Wadhwa, L. Bherer, M. S. Peterson, P. E. Scalf, J. S. Kim, M. Alvarado and A. F. Kramer (2007). "Training-induced plasticity in older adults: effects of training on hemispheric asymmetry." Neurobiol Aging 28(2): 272-283.
Grady, C. (2012). "The cognitive neuroscience of ageing." Nat Rev Neurosci 13(7): 491-505.
Hamacher, D., F. Herold, P. Wiegel, D. Hamacher and L. Schega (2015). "Brain activity during walking: A systematic review." Neurosci Biobehav Rev 57: 310-327.
Jancke, L., N. J. Shah and M. Peters (2000). "Cortical activations in primary and secondary motor areas for complex bimanual movements in professional pianists." Brain Res Cogn Brain Res 10(1-2): 177-183.
Kennedy, K. M., K. M. Rodrigue, G. N. Bischof, A. C. Hebrank, P. A. Reuter-Lorenz and D. C. Park (2015). "Age trajectories of functional activation under conditions of low and high processing demands: an adult lifespan fMRI study of the aging brain." Neuroimage 104: 21-34.
la Fougere, C., A. Zwergal, A. Rominger, S. Forster, G. Fesl, M. Dieterich, T. Brandt, M. Strupp, P. Bartenstein and K. Jahn (2010). "Real versus imagined locomotion: a [18F]-FDG PET-fMRI comparison." Neuroimage 50(4): 1589-1598.
Lustig, C., P. Shah, R. Seidler and P. A. Reuter-Lorenz (2009). "Aging, training, and the brain: a review and future directions." Neuropsychol Rev 19(4): 504-522.
Naito, E. and S. Hirose (2014). "Efficient foot motor control by Neymar's brain." Front Hum Neurosci 8: 594.
Raz, N. and K. M. Kennedy (2009). A systems approach to the aging brain: Neuroanatomic changes, their modifiers, and cognitive correlates. Imaging the Aging Brain. W. Jagust and M. D'Esposito. New York, NY, Oxford University Press: 43-70.
Reuter-Lorenz, P. A. and K. A. Cappell (2008). "Neurocognitive aging and the compensation hypothesis." Current Directions in Psychological Science 17(3): 177-182.
Stern, Y. (2009). "Cognitive reserve." Neuropsychologia 47(10): 2015-2028.
Tamnes, C. K., K. B. Walhovd, A. M. Dale, Y. Ostby, H. Grydeland, G. Richardson, L. T. Westlye, J. C. Roddey, D. J. Hagler, Jr., P. Due-Tonnessen, D. Holland, A. M. Fjell and I. Alzheimer's Disease Neuroimaging (2013). "Brain development and aging: overlapping and unique patterns of change." Neuroimage 68: 63-74.
West, R. L. (1996). "An application of prefrontal cortex function theory to cognitive aging." Psychol Bull 120(2): 272-292.
Zwergal, A., J. Linn, G. Xiong, T. Brandt, M. Strupp and K. Jahn (2012). "Aging of human supraspinal locomotor and postural control in fMRI." Neurobiol Aging 33(6): 1073-1084.

Experimental design

This is an important and methodologically well performed systematic review on an important emerging research topic that will be of interest to the readers of this journal. Overall, this review of the relevant literature is thorough.

Validity of the findings

A systematic review should not only provide readers a quick way to determine which papers they can trust based on for example a quality assessment, however, they should also offer readers contextual information what those papers are telling us by interpreting the findings and putting these in a context of existing works. They allow us to get a big picture of what is known in a field by doing this.
I do not see obvious issues with methodology or conclusions, but offer a few suggestions in hopes that the authors might find them helpful in revising their manuscript.

Annotated reviews are not available for download in order to protect the identity of reviewers who chose to remain anonymous.

---

## Round 0.2 · accepted · Accept

Thank you again for your careful attention to the issues raised in review. I believe that the reviewers' comments, along with your revisions, have improved an already very good piece of work, and I look forward to it's final publication.

# Reviewer 3 ·

Basic reporting

No comment

Experimental design

No comment

Validity of the findings

No comment

Additional comments

The authors did a nice job of revising their manuscript.
In Table 4: Write “Nieuwhof” not “Nieuwholf”